# LAMDA: A Longitudinal Android Malware Benchmark for Concept Drift Analysis

**Md Ahsanul Haque**[1]**, Ismail Hossain**[1]**, Md Mahmuduzzaman Kamol**[1]**, Md Jahangir Alam**[1]**,
Suresh kumar Amalapuram**[2]**, Sajedul Talukder**[1]**, Mohammad Saidur Rahman**[1]
[1]Department of Computer Science, University of Texas at El Paso
[2]Indian Institute of Technology Hyderabad,
{mhaque3,ihossain,mkamol,malam10}@miners.utep.edu
apskumarkrc@gmail.com, {stalukder,msrahman3}@utep.edu

## Abstract

Machine learning (ML)-based malware detection systems often fail to account for the dynamic nature of real-world training and test data distributions. In practice, these distributions evolve due to frequent changes in the Android ecosystem, adversarial development of new malware families, and the continuous emergence of both benign and malicious applications. Prior studies have shown that such concept drift—distributional shifts in benign and malicious samples, leads to significant degradation in detection performance over time. Despite the practical importance of this issue, existing datasets are often outdated and limited in temporal scope, diversity of malware families, and sample scale, making them insufficient for the systematic evaluation of concept drift in malware detection.

To address this gap, we present LAMDA, the largest and most temporally diverse Android malware benchmark to date, designed specifically for concept drift analysis. LAMDA spans 12 years (2013–2025, excluding 2015), includes over 1 million samples (approximately 37% labeled as malware), and covers 1,380 malware families and 150,000 singleton samples, reflecting the natural distribution and evolution of real-world Android applications. We empirically demonstrate LAMDA's utility by quantifying performance degradation of standard ML models over time and analyzing feature stability across years. As the most comprehensive Android malware dataset to date, LAMDA enables in-depth research into temporal drift, generalization, explainability, and evolving detection challenges.

The dataset and code are available at: https://iqsec-lab.github.io/LAMDA/.

## 1 Introduction

Android malware poses a growing threat to user privacy and security, with over 33 million attacks blocked in 2024 alone (Kaspersky, 2024; AV-TEST, 2025). Static feature based ML methods, which analyze features extracted from Android application packages (APKs), have emerged as a promising defense mechanism (Arp et al., 2014; Mariconti et al., 2017). However, these detectors often suffer performance degradation over time due to *concept drift* — gradual shifts in the feature distribution caused by the evolving nature of both malicious and benign software (Yang et al., 2021b).

Concept drift can result from several factors, including changes in developer practices, updates to Android APIs, and, most significantly, the evolving and adaptive strategies of malware authors (Greenberg, 2020). To evade detection, adversaries frequently obfuscate or modify their code by injecting alternative API calls, altering manifest components, or exploiting newly introduced services (News, 2024). For example, the Android trojan *SoumniBot* obfuscates its manifest file to evade analysis and detection (News, 2024). These tactics lead to observable shifts in static features over time, undermining the robustness of ML-based detection systems (Yang et al., 2021b). Prior studies have shown that malware families (i.e., clusters of samples exhibiting similar behavioral traits) play a central role in driving such drifts (Chow et al., 2023; Barbero et al., 2022).

Although concept drift plays a central role in Android malware evolution, most existing datasets are not designed to support drift analysis. Datasets such as Drebin (Arp et al., 2014), TESSER-

ACT (Pendlebury et al., 2019), and APIGraph (Zhang et al., 2020) are limited in temporal coverage, family diversity, or structural organization for studying drift. Similarly, Windows-based datasets like EMBER (Anderson & Roth, 2018), SOREL-20M (Harang & Rudd, 2020), and BODMAS (Yang et al., 2021a) are constrained by short collection periods or target different ecosystems. While EMBERSim (Corlatescu et al., 2023), MalNet (Freitas et al., 2022), and AnoShift (Dragoi et al., 2022) offer task-specific contributions, they do not support longitudinal drift analysis in Android malware classification. To address these gaps, we introduce LAMDA, a novel Android malware benchmark dataset curated for temporal drift analysis with family evolution. LAMDA spans over 12 years (i.e., 2013–2025, excluding 2015 due to the unavailability of hashes in the AndroZoo repository (Allix et al., 2016)), covering 1,008,381 APK samples across 1,380 unique malware families and over 150,000 Singleton samples (i..e, samples without *av class* labels) from AndroZoo repository (Alecci et al., 2024). Each sample is labeled using VirusTotal's `vt_detection` count (VirusTotal, 2025) reported in AndroZoo database (Alecci et al., 2024). The samples are decompiled to extract fine-grained static features following the Drebin (Arp et al., 2014) feature definitions.

We validate LAMDA through a series of comprehensive evaluations, including longitudinal degradation analysis of the supervised binary classification under concept drift (AnoShift-style (Dragoi et al., 2022)), temporally disjoint training (testing), and family-wise feature stability assessments. LAMDA enables explanation-guided analysis of concept drift and combines long-term structural modeling with SHAP-based attributions (Lundberg & Lee, 2017), allowing researchers to trace how feature relevance shifts over time and better understand the underlying causes of model degradation.

In summary, the contributions of this paper are as follows:

- We present LAMDA, a large-scale Android malware benchmark comprising over 1 million APKs across 1,380 unique families spanning for 12 years (2013 to 2025, excluding 2015), built on static features based on Drebin (Arp et al., 2014) features.
- We perform a detail concept drift detection using structured temporal splits (Dragoi et al., 2022) to show that LAMDA exhibits pronounced distributional shift than prior benchmark.
- We conduct comprehensive drift analysis, including per-feature distribution shifts, feature stability analysis across malware families (Zhang et al., 2020), temporal label flipping analysis, and SHAP-based explanation (Lundberg & Lee, 2017) drift that reveal temporal changes in feature importance.
- We show that existing drift adaptation methods, though effective on prior benchmark (Zhang et al., 2020), fail to generalize on LAMDA due to its more realistic and pronounced concept drift.

## 2 RELATED WORK

In this section, we discuss prior work and their limitations that motivate the creation of LAMDA.

**Evolution of Malware Datasets and Benchmarks.** Early malware datasets such as Drebin (Arp et al., 2014) (Android) and EMBER (Anderson & Roth, 2018) (Windows) have played a pivotal role to study concept drift in malware analysis. More recent efforts—including SOREL-20M (Harang & Rudd, 2020) and BODMAS (Yang et al., 2021a) for Windows, and TESSERACT (Pendlebury et al., 2019), APIGraph (Zhang et al., 2020), and Chen et al. (2023) for Android attempt to address limitations in scale and recency. Nonetheless, these datasets suffer from one or more major limitations — they are often outdated, contain either relatively few malware samples or families, or lack long-term temporal coverage necessary for studying the evolution of malware. For example, Drebin spans only 2010–2012 with 5,560 samples from 179 families; TESSERACT covers 2014–2016 with 12,735 samples; API Graph spans 2012–2018 with 32,089 samples from 1,120 families; and Chen et al. (2023) includes 10,200 samples across 254 families from 2019–2021. Despite their temporal spread, these datasets are not explicitly structured to support longitudinal drift analysis or capture evolutionary patterns in malware behavior.

**Explainability and Semantic Features.** Explainability is critical for understanding how feature importance shifts under concept drift. While Drebin (Arp et al., 2014) and BODMAS (Yang et al., 2021a) introduced interpretable features and temporal structure, few studies have systematically used them to analyze drift. TRANSCENDENT (Barbero et al., 2022) incorporates semantic reasoning for selective prediction, but longitudinal robustness of explanations remains underexplored due

to limited dataset support. LAMDA fills this gap by providing a temporally structured benchmark with interpretable features and SHapley Additive exPlanations (SHAP)-based explanations (Lundberg & Lee, 2017), enabling fine-grained, longitudinal analysis of model behavior and drift.

# 3 LAMDA CREATION

In this section, we describe the construction process of the LAMDA. We have downloaded APKs from AndrooZoo repository (Allix et al., 2016) and decompiled APKs to extract static Drebin (Arp et al., 2014) features and then transformed the features into binary vectors for downstream analysis.

**Label Assignment and Collection Strategy.** To construct a large-scale, temporally diverse dataset, we use metadata from AndroZoo (Allix et al., 2016), including APK hashes, VirusTotal (VT) results, and submission dates. For each year from 2013 to 2025 (excluding 2015, which lacks valid entries), we collect APKs and assign binary labels using the `vt_detection` field. Following prior heuristics (Arp et al., 2014; Pendlebury et al., 2019), we define: (i) *Benign* for `vt_detection` $= 0$, (ii) *Malware* for `vt_detection` $\geq 4$, and (iii) discard *Uncertain* samples with scores in $1, 3$. The $\geq 4$ threshold mitigates label noise by requiring stronger AV consensus (Chen et al., 2016).

To reduce sampling bias in learning systems, we collected 50,000 malware and 50,000 benign samples per year, while preserving month-wise temporal distributions across both categories. Although prior work such as TESSERACT (Pendlebury et al., 2019; Chen et al., 2023) adopts a 90:10 benign-to-malware ratio, we attempt to maintain a balanced 50:50 ratio (Anderson & Roth, 2018). This choice is motivated by the need to mitigate the risk of skewed learned representations (such as overfitting (Shwartz-Ziv et al., 2023), disparity in learning (Zhou et al., 2023)) that can arise from class imbalance. A balanced dataset helps ensure that the model learns meaningful distinctions between classes, captures a wider range of malware families, and is exposed to a broader spectrum of behaviors and evasive techniques. Such diversity not only enables longitudinal generalization studies but also increases the difficulty of the detection task, particularly for learning systems that must contend with rare, novel, or semantically similar malware families (Anderson & Roth, 2018). Nonetheless, due to limited availability of malware samples in certain years such as 2017, 2023, 2024, and 2025, LAMDA still exhibits class imbalance, with each of these years showing different imbalance ratios.

Another practical challenge during data collection involved download and decompilation failures, requiring us to over-fetch APKs to meet target counts. To mitigate this, we included a 20% overhead in the number of APK hashes per year. All APKs are retrieved via authenticated academic access to the AndroZoo repository [1] and stored in a consistent directory structure (`[year]/malware/`, `[year]/benign/`) to facilitate temporal slicing and cross-year analysis. Corrupted or undecompilable samples are excluded and logged for transparency. The final dataset comprises over one million APKs. A detailed year-wise breakdown is provided in Appendix A.

**Label Assignment.** We utilized the original labels provided by AndroZoo (Allix et al., 2016) and their VirusTotal scan dates range from 2009 to 2025, depending on the APKs. Due the original label scan date variation, we re-scanned all malware samples in our dataset using VirusTotal at the time of data collection. We reported the label drift observed between the original AndroZoo labels and our re-scan labels in Section 5.4.

**Family Label Acquisition.** To enable finer-grained analysis of how malware behavior evolves over time, we assign family-level labels to all malware samples using `AVClass2` (Sebastián et al., 2016; Sebastián & Caballero, 2020), which standardizes noisy antivirus vendor labels into consistent malware family names. This is critical for developing detection systems that generalize to emerging threats. The labeling process involves retrieving VirusTotal (2025) reports, converting them to the required format, running AVClass2, and post-processing the output to retain SHA256 mappings. Figures 1(a) and 1(b) show the yearly distribution of recurring versus newly observed families and the count of singleton families—those that appear only once—respectively. Family labels enable research into more complex tasks such as multi-class classification and the study of temporal trends across malware families.

---

[1] https://androzoo.uni.lu/access

Table 1: APIGraph and LAMDA Dataset Statistics.

| Year | APIGraph | | | | LAMDA | | | |
|------|----------|---------|--------|-----------|--------|---------|--------|-----------|
| | Benign | Malware | Family | Singleton | Benign | Malware | Family | Singleton |
| 2012 | 27613 | 3066 | 104 | 36 | - | - | - | - |
| 2013 | 43873 | 4871 | 172 | 68 | 42048 | 44383 | 213 | 1550 |
| 2014 | 52843 | 5871 | 175 | 55 | 55427 | 45756 | 231 | 2482 |
| 2015 | 52173 | 5797 | 193 | 53 | - | - | - | - |
| 2016 | 50859 | 5651 | 199 | 68 | 64059 | 45134 | 375 | 5861 |
| 2017 | 24930 | 2620 | 147 | 48 | 77785 | 21359 | 207 | 9063 |
| 2018 | 38214 | 4213 | 128 | 45 | 64942 | 39350 | 373 | 20579 |
| 2019 | - | - | - | - | 49465 | 41585 | 635 | 18916 |
| 2020 | - | - | - | - | 55718 | 46355 | 588 | 30644 |
| 2021 | - | - | - | - | 45528 | 35627 | 295 | 30020 |
| 2022 | - | - | - | - | 44768 | 41648 | 651 | 24927 |
| 2023 | - | - | - | - | 46462 | 7892 | 224 | 5922 |
| 2024 | - | - | - | - | 47633 | 794 | 64 | 626 |
| 2025 | - | - | - | - | 44640 | 23 | 8 | 14 |
| Total | 290,505 | 32,089 | | 373 | 638,475 | 369,906 | | 150,604 |

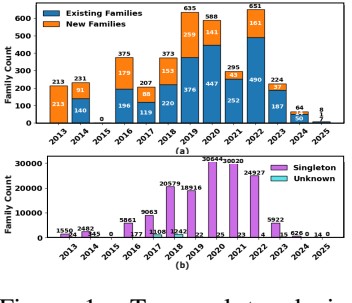

Figure 1: Temporal trends in malware family evolution.

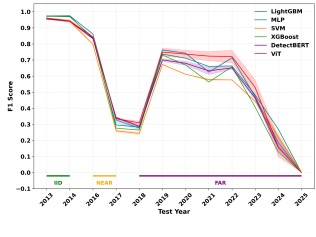

(a) LAMDA.

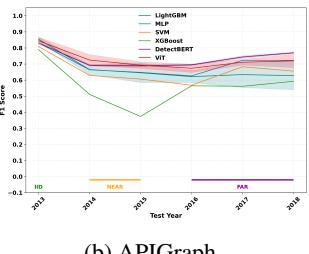

(b) APIGraph.

Figure 2: F1-score over time across different temporal splits.

**Decompilation and Static Feature Extraction.** Each APK is statically decompiled using `apktool` (Brut, 2025), producing a disassembled `smali` representation and the original `AndroidManifest.xml`. We parse these artifacts to extract a diverse set of static features commonly used in Android malware detection (Arp et al., 2014). Specifically, the `AndroidManifest.xml` file is analyzed to obtain the list of requested permissions (e.g., `ACCESS_FINE_LOCATION`), declared activities and services, broadcast receivers, required hardware components, and intent filters (Arp et al., 2014). Meanwhile, the disassembled `smali` code is scanned to identify invocations of restricted APIs (e.g., `NotificationManager.notify`), suspicious API usages (e.g., `getSystemService`), and embedded hardcoded IPs/URLs (e.g., `e.crashlytics.com`). The extracted Drebin feature sets comprises several static categories derived from Android APKs (Arp et al., 2014). A detailed list of features is provided in Appendix B.

**Vectorization and Temporal Feature Alignment.** After decompiling each APK, we extract static features into a `.data` file. Each year's data is split into 80% training and 20% testing sets using stratified sampling to preserve class balance. From the training set, we construct a global vocabulary by taking the union of unique tokens across all samples, yielding 9,690,482 ($\approx 9.69$ million) raw features (Yang et al., 2021b). Each APK is then represented as a high-dimensional binary vector using a bag-of-tokens model, where each token corresponds to a binary feature indicating its presence or absence in the sample (Arp et al., 2014). To reduce dimensionality and ensure computational feasibility, we apply `VarianceThreshold` from `scikit-learn` to eliminate low-variance features. For all experiments, we use the `Baseline` variant, which applies a threshold of 0.001 (Rahman et al., 2025), resulting in $4,561$ final features. This compact and consistent representation supports a range of downstream tasks, including supervised learning, drift analysis, and continual learning. More details are in Appendix B. The dataset is initially created in a sparse matrix format, storing binary feature vectors and metadata as compressed `.npz` files to optimize for storage and computational efficiency. These `.npz` files are organized by year and stratified into *training* and *test* splits. A detailed breakdown of feature dimensions under varying `VarianceThreshold` settings is provided in Appendix B. For scalability of LAMDA, we have also published global features, variance threshold objects and selected features after applying `VarianceThreshold`.

**Tabular Comparison of LAMDA and APIGraph.** Table 1 provides the statistics of API-Graph Zhang et al. (2020) and LAMDA in terms of number of benign, malware, families and singleton samples for each year.

# 4 CONCEPT DRIFT DETECTION

In this section, we examine performance degradation of supervised models across temporally distant splits (Dragoi et al., 2022) (Section 4.1) to detect concept drift, followed by distributional shifts using Jeffreys divergence and t-SNE visualizations (Sections 4.1 and 4.2).

## 4.1 CONCEPT DRIFT DETECTION WITH SUPERVISED LEARNING

**Experimental Setting.** To evaluate the robustness of malware detectors under temporal distribution shifts, we perform supervised learning experiments using four widely adapted detector models from the malware research — Linear SVM, LightGBM, MLP, XGBoost, detectBERT, and ViT (Arp et al., 2014; Anderson & Roth, 2018). Detailed model configurations are provided in Appendix D.

Following the AnoShift benchmark (Dragoi et al., 2022), we divide LAMDA into three temporally separated regions: TRAIN (i.e., initial training set), IID, NEAR, and FAR. TRAIN+IID includes samples from 2013–2014, with the last month of each year held out for IID evaluation. Models are trained on all other months, and the held-out portion serves as an in-distribution test set to measure baseline performance on temporally adjacent, unseen data. To assess generalization under drift, we define two evaluation regions: NEAR (2016–2017) and FAR (2018–2025), allowing systematic analysis of performance degradation as the temporal gap from training increases. For comparison, we evaluate the same models on the APIGraph dataset (Zhang et al., 2020) using a similar split: training on 2012, IID on 2013, NEAR on 2014, and FAR on 2015–2018. All experiments are repeated five times with different random seeds. For each split, we report results in Table 2 as *mean±std*, averaged across all runs and years within each split. Figure 2 shows yearly results averaged over runs. This experiment uses the LAMDA-baseline with a `VarianceThreshold` of 0.001. To study the impact of feature space on drift, we also test thresholds of 0.01 and 0.0001, with results provided in Appendix D.

**Results.** Table 2 summarizes the performance of malware detectors on both LAMDA and API-Graph under the IID, NEAR, and FAR evaluation splits. All detectors perform strongly under IID conditions, but their effectiveness declines sharply as the temporal gap from training increases. For instance, LightGBM's F1-score on LAMDA drops from 97.49% (IID) to 59.48% (NEAR) and 47.24% (FAR), alongside a significant rise in the false negative rate, from 1.47% to 50.51% and 64.10%, respectively,—demonstrating increased difficulty. In contrast, the false positive rate (FPR) remains low and stable, likely due to the more consistent behavior of benign apps over time. Figure 2(a) further visualizes this trend, showing how F1-scores decline over time. Notably, we observe a sharp drop in performance between 2016 and 2017, indicating a significant distributional shift. A similar decline is evident from 2023 to 2024. In contrast, F1-scores increase from 2018 to the 2019–2022 period, suggesting that these intermediate years exhibit less drift relative to 2017 and 2018.

In APIGraph, LightGBM's F1-score drops from 85.95% (IID) to 66.77% (NEAR), but stabilizes at 68.20% on FAR. The F1-scores over the years in Figure 2(b) indicate a smaller degree of temporal drift, with only modest changes between years. Compared to the APIGraph, LAMDA shows a higher standard deviation in both NEAR and FAR, suggesting more pronounced and variable distributional shifts. Complex transformer-based models like DetectBERT and ViT also degrade with increasing drift severity. This supports our claim that LAMDA introduces stronger concept drift, making it a more challenging and realistic benchmark for evaluating long-term malware detection.

**Significant Performance Drop in 2017 and 2018.** Figure 2(a) shows a significant performance drop in 2017 and 2018 for LAMDA. This drop also aligns with multiple independent indicators of strong drift present in these two years. Figure 3 shows a sharp increase in Jeffrey's divergence between 2016 to 2017 and from 2017 to 2018, indicating substantial shifts in static features such as API usage and permissions. Feature stability results in Figure 6a highlight that features are most unstable in 2017–2018, with larger fluctuations across malware families, suggesting significant

Table 2: Comparison of performances on LAMDA and API Graph across three temporal splits.

| Split | Model | LAMDA | | | | | API Graph | | | | |
|---|---|---|---|---|---|---|---|---|---|---|---|
| | | F1 | ROC-AUC | PR-AUC | FNR | FPR | F1 | ROC-AUC | PR-AUC | FNR | FPR |
| IID | LightGBM | $97.49_{\pm0.17}$ | $99.55_{\pm0.03}$ | $99.50_{\pm0.11}$ | $1.74_{\pm0.34}$ | $2.69_{\pm0.48}$ | $85.95_{\pm0.00}$ | $98.91_{\pm0.00}$ | $95.20_{\pm0.00}$ | $22.39_{\pm0.00}$ | $0.33_{\pm0.00}$ |
| | MLP | $97.21_{\pm0.12}$ | $99.48_{\pm0.04}$ | $99.38_{\pm0.20}$ | $2.50_{\pm0.06}$ | $2.58_{\pm0.85}$ | $85.79_{\pm0.00}$ | $96.37_{\pm0.00}$ | $88.49_{\pm0.00}$ | $20.31_{\pm0.00}$ | $0.67_{\pm0.00}$ |
| | SVM | $94.98_{\pm1.07}$ | $98.89_{\pm0.28}$ | $98.75_{\pm0.46}$ | $4.82_{\pm0.76}$ | $4.09_{\pm0.55}$ | $82.00_{\pm0.00}$ | $97.33_{\pm0.00}$ | $90.94_{\pm0.00}$ | $26.74_{\pm0.00}$ | $0.60_{\pm0.00}$ |
| | XGBoost | $97.05_{\pm0.14}$ | $99.15_{\pm0.16}$ | $97.68_{\pm1.16}$ | $2.20_{\pm0.43}$ | $2.96_{\pm0.01}$ | $80.33_{\pm0.00}$ | $96.05_{\pm0.00}$ | $89.74_{\pm0.00}$ | $28.35_{\pm0.00}$ | $0.75_{\pm0.00}$ |
| | DetectBERT | $95.27_{\pm0.86}$ | $98.92_{\pm0.27}$ | $98.61_{\pm0.79}$ | $3.96_{\pm0.13}$ | $4.48_{\pm0.59}$ | $83.05_{\pm0.00}$ | $98.82_{\pm0.00}$ | $94.19_{\pm0.00}$ | $25.60_{\pm0.00}$ | $0.48_{\pm0.00}$ |
| | ViT | $94.97_{\pm1.59}$ | $98.91_{\pm0.37}$ | $98.61_{\pm0.85}$ | $4.63_{\pm0.81}$ | $4.17_{\pm0.73}$ | $86.64_{\pm0.00}$ | $98.65_{\pm0.00}$ | $93.73_{\pm0.00}$ | $17.82_{\pm0.00}$ | $0.81_{\pm0.00}$ |
| NEAR | LightGBM | $59.48_{\pm28.20}$ | $74.05_{\pm23.76}$ | $70.18_{\pm27.10}$ | $50.51_{\pm30.82}$ | $1.85_{\pm0.95}$ | $66.77_{\pm0.00}$ | $95.94_{\pm0.00}$ | $83.01_{\pm0.00}$ | $47.68_{\pm0.00}$ | $0.48_{\pm0.00}$ |
| | MLP | $56.57_{\pm28.41}$ | $82.71_{\pm11.19}$ | $67.94_{\pm24.59}$ | $51.95_{\pm30.42}$ | $3.98_{\pm1.19}$ | $68.72_{\pm0.00}$ | $86.79_{\pm0.00}$ | $68.31_{\pm0.00}$ | $38.80_{\pm0.00}$ | $1.87_{\pm0.00}$ |
| | SVM | $52.91_{\pm28.40}$ | $75.18_{\pm17.98}$ | $62.53_{\pm29.82}$ | $55.62_{\pm28.88}$ | $4.71_{\pm0.97}$ | $63.06_{\pm0.00}$ | $89.25_{\pm0.00}$ | $70.15_{\pm0.00}$ | $45.48_{\pm0.00}$ | $2.03_{\pm0.00}$ |
| | XGBoost | $55.84_{\pm29.73}$ | $77.75_{\pm16.85}$ | $68.14_{\pm26.55}$ | $53.84_{\pm30.94}$ | $2.14_{\pm0.86}$ | $51.47_{\pm0.00}$ | $86.90_{\pm0.00}$ | $65.69_{\pm0.00}$ | $60.41_{\pm0.00}$ | $1.57_{\pm0.00}$ |
| | DetectBERT | $59.11_{\pm34.70}$ | $73.86_{\pm29.73}$ | $66.95_{\pm37.60}$ | $49.92_{\pm37.55}$ | $4.05_{\pm0.84}$ | $69.08_{\pm0.01}$ | $94.33_{\pm0.45}$ | $82.54_{\pm1.20}$ | $43.13_{\pm0.71}$ | $0.59_{\pm0.15}$ |
| | ViT | $58.77_{\pm35.48}$ | $71.47_{\pm32.31}$ | $63.97_{\pm40.01}$ | $48.68_{\pm39.01}$ | $5.63_{\pm1.28}$ | $72.15_{\pm0.00}$ | $93.08_{\pm0.00}$ | $78.26_{\pm0.00}$ | $31.23_{\pm0.00}$ | $2.41_{\pm0.00}$ |
| FAR | LightGBM | $47.24_{\pm27.33}$ | $78.04_{\pm20.83}$ | $63.45_{\pm35.80}$ | $64.10_{\pm22.97}$ | $1.30_{\pm0.95}$ | $68.20_{\pm4.63}$ | $95.68_{\pm0.81}$ | $81.69_{\pm1.59}$ | $45.04_{\pm5.72}$ | $0.61_{\pm0.07}$ |
| | MLP | $47.59_{\pm25.30}$ | $84.04_{\pm11.23}$ | $66.16_{\pm34.34}$ | $64.40_{\pm20.63}$ | $1.14_{\pm0.71}$ | $63.92_{\pm5.39}$ | $87.10_{\pm1.95}$ | $64.22_{\pm4.19}$ | $46.45_{\pm6.10}$ | $1.40_{\pm0.15}$ |
| | SVM | $41.86_{\pm22.55}$ | $79.07_{\pm15.06}$ | $62.27_{\pm34.09}$ | $69.93_{\pm16.85}$ | $1.27_{\pm0.76}$ | $66.18_{\pm6.21}$ | $93.44_{\pm0.52}$ | $75.46_{\pm2.58}$ | $44.26_{\pm6.35}$ | $1.23_{\pm0.14}$ |
| | XGBoost | $42.75_{\pm25.86}$ | $76.85_{\pm16.49}$ | $60.33_{\pm35.88}$ | $68.11_{\pm20.43}$ | $1.69_{\pm0.57}$ | $54.88_{\pm9.26}$ | $83.44_{\pm4.79}$ | $65.03_{\pm6.67}$ | $56.68_{\pm8.92}$ | $1.41_{\pm0.35}$ |
| | DetectBERT | $45.01_{\pm26.84}$ | $76.37_{\pm23.07}$ | $60.56_{\pm36.63}$ | $64.57_{\pm20.54}$ | $3.15_{\pm1.09}$ | $73.23_{\pm3.68}$ | $96.74_{\pm0.17}$ | $83.15_{\pm1.44}$ | $35.67_{\pm5.55}$ | $0.99_{\pm0.15}$ |
| | ViT | $47.03_{\pm29.55}$ | $78.53_{\pm20.17}$ | $58.77_{\pm37.82}$ | $60.23_{\pm22.66}$ | $4.77_{\pm1.20}$ | $68.47_{\pm3.94}$ | $95.28_{\pm0.19}$ | $73.69_{\pm3.83}$ | $27.71_{\pm6.81}$ | $3.81_{\pm0.14}$ |

(a) LAMDA  (b) APIGraph

Figure 3: Jeffreys divergence heatmaps across years for LAMDA and APIGraph.

Figure 4: t-SNE projections showing feature space evolution for LAMDA and APIGraph.

behavioral changes. SHAP-based explanation drift (Figure 7a) also shows a strong drop in 2017 and 2018, confirming abrupt changes in the model's decision logic during this period.

## 4.2 VISUAL ANALYSIS OF CONCEPT DRIFT

**Setting.** To track how malware and benign class distributions evolve over time, we use two complementary visualization techniques: Jeffreys divergence heatmaps and t-SNE projections. Jeffreys divergence (Jeffreys, 1946), a symmetric information-theoretic measure, quantifies distributional shifts in individual static features across years. We compute pairwise divergences for all yearly combinations in LAMDA (2013–2025) and APIGraph (2012–2018). In parallel, we apply t-SNE (Van der Maaten & Hinton, 2008) to project high-dimensional feature vectors into 2D space. For direct comparison, we focus on four years shared by both datasets (2013, 2014, 2016, 2017), following common practice in prior malware drift studies (Pendlebury et al., 2019). Full year-wise t-SNE visualizations are provided in Appendix B.

**Analysis.** Figure 3 shows Jeffreys divergence heatmaps for both datasets. In both LAMDA and APIGraph, divergence increases as the temporal gap widens, confirming non-trivial concept drift. LAMDA exhibits a broader divergence range, particularly from 2022–2025, reflecting substantial feature distribution changes likely driven by evolving APIs, shifting development practices, and emerging malware behaviors. In contrast, APIGraph remains relatively stable, with limited divergence in its later years. Figure 4 provides t-SNE projections for the selected years. In LAMDA, samples become more dispersed by 2016–2017, suggesting increasing structural sparsity in feature space. Since t-SNE distorts global distances, we corroborate these patterns with Jeffreys divergence, which confirms genuine distributional shifts. APIGraph, by comparison, maintains tight and relatively static clusters, indicating limited structural evolution. Together, these trends underscore LAMDA's value as a temporally rich benchmark for studying real-world concept drift.

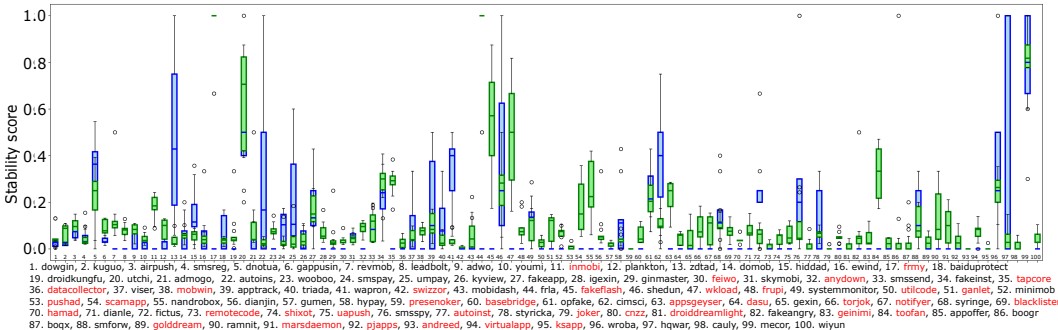

Figure 5: The distribution of feature stability scores for top 100 malware families. 58 families are common in both LAMDA (**green**) and APIGraph (**blue**) datasets, and families marked as red labels along $x$-axis available in LAMDA with minimum family size criteria.

## 5 COMPREHENSIVE DRIFT ANALYSIS

In this section, we present a more in-depth analysis of different types of drift, including feature stability (Section 5.1), temporal drift (Section 5.2), SHAP-based explainability (Section 5.3), and label drift (Section 5.4)

### 5.1 FEATURE SPACE STABILITY ANALYSIS ON TOP MALWARE FAMILIES

**Analysis Setting.** We evaluate the temporal consistency of malware families using two complementary metrics: stability scores and Optimal Transport Dataset Distance (OTDD) (Alvarez-Melis & Fusi, 2020), following prior work (Zhang et al., 2020; Dragoi et al., 2022). The analysis is conducted in the original feature space and focuses on the 100 malware families with the largest sample sizes. Within each family, samples are temporally ordered from 2013 to 2025, though not all families contain data for every year. Following (Zhang et al., 2020), we partition each family's samples into ten equal subsets, each representing 10% of the family's total. For APIGraph, we identify 58 families with at least 10 samples between 2013 and 2018, meeting this subdivision requirement. Stability scores are then computed using Jaccard similarity across the ten subsets for both LAMDA and APIGraph.

**Stability Scores Analysis.** Figure 5 shows the distribution of consecutive pairwise stability scores across ten groups for each of the top 100 malware families. The number of samples per family varies considerably, ranging from 186 to 32,475, with a mean of 1,984 and a median of 535. The green box plots correspond to LAMDA, while the blue box plots represent APIGraph. Both datasets capture the temporal evolution of malware families, as reflected in the spread and median of stability scores. Broader spreads and lower medians in both datasets indicate greater behavioral variability over time. Notably, LAMDA includes more families and reflects broader evolutionary patterns than APIGraph. These differences suggest that detection models trained on LAMDA may offer improved insight into concept drift, benefiting from greater sample diversity and family coverage.

### 5.2 TEMPORAL DRIFT ANALYSIS ON COMMON MALWARE FAMILIES

**Analysis Setting.** We assess the drifting behavior over the years for the common families present from 2013 to 2025. We observe that *only 10* families appear consistently each year, except for 2025. Subsequently, we compute the year-wise stability score for the original feature set within each of these 10 common families. Additionally, we measure the distribution distances based on the CADE (Yang et al., 2021b) latent features in the test set. This experiment uses 2013 dataset for training and 2014 to 2024 samples serve as test sets.

**Feature-Based Stability Evaluation.** Figure 6a shows Jaccard similarity–based stability scores across consecutive yearly sample sets for 10 common malware families. Flatter curves indicate stronger temporal consistency, while sharp variations reflect feature drift. Most families such as `airpush`, `dianjin`, and `smsreg` exhibit relatively stable trends, suggesting consistent feature

Table 3: This table shows label drift for Android malware samples, highlighting shifts in detection over time. **TS**: Total # of Malware Samples, $B_C$: Currently Labeled as Benign, $\%B_C$: Percentage of Total Malware Samples Currently Labeled as Benign. $D_{Improved}$: Improved Detection, $D_{Weakened}$: Weakened Detection, $D_{Unchanged}$: Unchanged Detection, $DS_{Drop}$: Significant Drop of Detection Count, $DS_{Improve}$: Detection Count Significantly Increased.

| Year | TS | $B_C$ | $\%B_C$ | $D_{Improved}$ | $D_{Weakened}$ | $D_{Unchanged}$ | $DS_{Drop}$ | $DS_{Increase}$ |
|------|------|------|------|------|------|------|------|------|
| 2013 | 44383 | 24 | 0.05 | 40436 | 439 | 3484 | 85 | 34481 |
| 2014 | 45756 | 345 | 0.75 | 37108 | 1554 | 6749 | 863 | 27239 |
| 2016 | 45134 | 177 | 0.39 | 26963 | 7485 | 10509 | 1160 | 13581 |
| 2017 | 21359 | 1108 | 5.19 | 7765 | 10289 | 2197 | 5061 | 3362 |
| 2018 | 39350 | 1242 | 3.16 | 17561 | 15346 | 5201 | 7304 | 7600 |
| 2019 | 41585 | 22 | 0.05 | 22905 | 9294 | 9364 | 467 | 7518 |
| 2020 | 46355 | 25 | 0.05 | 20755 | 3931 | 21644 | 294 | 8001 |
| 2021 | 35627 | 23 | 0.06 | 10385 | 4482 | 20737 | 176 | 2531 |
| 2022 | 41648 | 4 | 0.01 | 10445 | 3629 | 27570 | 121 | 2719 |
| 2023 | 7892 | 15 | 0.19 | 1763 | 1979 | 4135 | 592 | 416 |
| 2024 | 794 | 0 | 0.00 | 74 | 319 | 401 | 79 | 19 |
| 2025 | 23 | 0 | 0.00 | 6 | 2 | 15 | 0 | 2 |

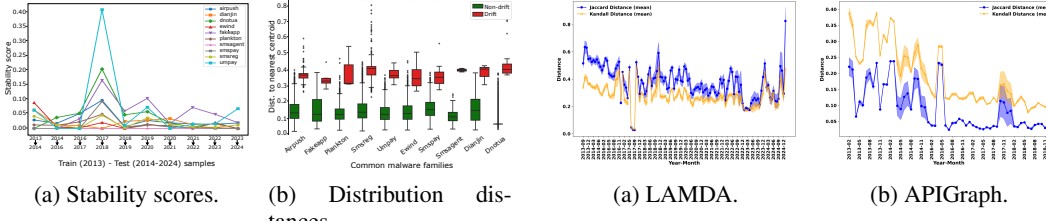

(a) Stability scores.    (b) Distribution distances.     (a) LAMDA.     (b) APIGraph.

Figure 6: Stability and distribution analysis on malware families.

Figure 7: SHAP-based explanation drift on LAMDA and APIGraph datasets.

distributions over time. In contrast, families like `umpay`, `fakeapp`, and `dnotua` display pronounced fluctuations, with a notable spike around 2017–2018. The especially large peak for `umpay` signals substantial temporal drift, likely tied to evolving malware behaviors during that period. Overall, these results show that while many families retain stable characteristics, others undergo significant shifts, underscoring the importance of dynamic adaptation in detection models.

**Latent Space Drift Detection via Distance Metrics.** Figure 6b shows the distribution of distances from test samples (2014–2024) to their nearest class centroids across 10 common families, computed in the contrastive latent space (Yang et al., 2021b). Each test sample is encoded using the trained contrastive autoencoder, and Euclidean distances to class centroids are computed and normalized within each class using the *Median Absolute Deviation* (MAD). A sample is classified as *drifted* if its normalized MAD score $A^{(k)}$ exceeds the empirical threshold $T_{\mathrm{MAD}} = 3.5$; otherwise, it is considered *non-drifted*. This criterion identifies samples that deviate significantly from learned class distributions as drift instances. The resulting boxplots reveal a clear separation between non-drifted (green) and drifted (red) samples across families. Drifted samples consistently exhibit higher distances, with more separation for families such as `plankton`, `umpay`, and `dianjin`. In contrast, non-drifted samples cluster tightly around their centroids, indicating intra-family stability. Additional analysis are provided in Appendix C.

## 5.3 Temporal Analysis of SHAP-based Explanation Drift

**Analysis Setting.** Explanation drift happens when the features a malware detector relies on for decisions change over time. To study this in the LAMDA and APIGraph datasets, we measure two types of change using SHAP (Lundberg & Lee, 2017) feature attributions: Jaccard distance, which measures overlap in important features across time, and Kendall distance, which measures ranking consistency. Small distances consistent reasoning, while large Jaccard or low Kendall values indicate shifts in reasoning. We compute SHAP values using `KernelExplainer` (Lundberg &

Table 4: Comparison of concept drift adaptation methods on LAMDA and APIGraph under different labeling budgets. Results are reported as mean$_{\pm\text{std}}$.

| Budget | Method | APIGraph | | | LAMDA | | |
|---|---|---|---|---|---|---|---|
| | | F1 (%) | FNR (%) | FPR (%) | F1 (%) | FNR (%) | FPR (%) |
| 50 | Chen et al. (2023) | $\mathbf{89.26}_{\pm 0.31}$ | $\mathbf{15.13}_{\pm 0.31}$ | $\mathbf{0.51}_{\pm 0.03}$ | $37.43_{\pm 2.43}$ | $\mathbf{3.54}_{\pm 0.32}$ | $93.79_{\pm 2.71}$ |
| | CADE Yang et al. (2021b) | $83.92_{\pm 1.75}$ | $19.79_{\pm 2.09}$ | $1.07_{\pm 0.15}$ | $34.20_{\pm 1.80}$ | $44.00_{\pm 2.50}$ | $\mathbf{56.80}_{\pm 2.20}$ |
| | TRANSCENDENT Barbero et al. (2022) | $38.42_{\pm 1.52}$ | $69.47_{\pm 1.83}$ | $1.65_{\pm 0.27}$ | $32.00_{\pm 1.50}$ | $37.80_{\pm 2.10}$ | $63.70_{\pm 1.80}$ |
| 100 | Chen et al. (2023) | $\mathbf{90.70}_{\pm 0.15}$ | $\mathbf{13.18}_{\pm 0.26}$ | $\mathbf{0.45}_{\pm 0.00}$ | $38.50_{\pm 2.20}$ | $\mathbf{3.30}_{\pm 0.30}$ | $93.50_{\pm 2.60}$ |
| | CADE Yang et al. (2021b) | $87.08_{\pm 0.54}$ | $14.39_{\pm 1.08}$ | $1.10_{\pm 0.16}$ | $37.30_{\pm 1.60}$ | $37.20_{\pm 2.10}$ | $\mathbf{45.20}_{\pm 2.00}$ |
| | TRANSCENDENT Barbero et al. (2022) | $40.17_{\pm 1.49}$ | $68.03_{\pm 1.67}$ | $1.42_{\pm 0.25}$ | $35.80_{\pm 1.30}$ | $28.40_{\pm 1.90}$ | $70.10_{\pm 1.60}$ |
| 200 | Chen et al. (2023) | $\mathbf{91.70}_{\pm 0.38}$ | $\mathbf{11.43}_{\pm 0.61}$ | $\mathbf{0.44}_{\pm 0.01}$ | $41.00_{\pm 1.80}$ | $\mathbf{2.85}_{\pm 0.28}$ | $91.20_{\pm 2.30}$ |
| | CADE Yang et al. (2021b) | $88.87_{\pm 0.24}$ | $14.05_{\pm 0.35}$ | $0.74_{\pm 0.01}$ | $38.50_{\pm 1.30}$ | $31.80_{\pm 1.80}$ | $\mathbf{38.50}_{\pm 1.60}$ |
| | TRANSCENDENT Barbero et al. (2022) | $42.38_{\pm 1.36}$ | $66.48_{\pm 1.54}$ | $1.15_{\pm 0.21}$ | $39.00_{\pm 1.20}$ | $21.30_{\pm 1.50}$ | $76.30_{\pm 1.40}$ |
| 400 | Chen et al. (2023) | $\mathbf{92.39}_{\pm 0.29}$ | $\mathbf{10.18}_{\pm 0.52}$ | $\mathbf{0.45}_{\pm 0.02}$ | $43.00_{\pm 1.60}$ | $\mathbf{2.50}_{\pm 0.25}$ | $89.80_{\pm 2.10}$ |
| | CADE Yang et al. (2021b) | $89.16_{\pm 0.53}$ | $13.23_{\pm 0.56}$ | $0.80_{\pm 0.15}$ | $\mathbf{45.40}_{\pm 1.10}$ | $59.20_{\pm 1.50}$ | $\mathbf{10.10}_{\pm 1.20}$ |
| | TRANSCENDENT Barbero et al. (2022) | $43.97_{\pm 1.21}$ | $65.41_{\pm 1.42}$ | $0.98_{\pm 0.17}$ | $40.60_{\pm 1.10}$ | $15.60_{\pm 1.30}$ | $82.80_{\pm 1.20}$ |

Lee, 2017) with 100 background and 100 test samples per month. We calculate distances over the top-1000 features. To make figures easier to read, the $x$-axis in Figure 7a is labeled every three months, covering June 2013–January 2025 (shown from September 2013–December 2024). For APIGraph, the period is January 2013–December 2018 with the same labeling scheme (Figure 7b).

**Jaccard and Kendall Distance Analysis.** Figure 7a shows that LAMDA exhibits consistently high Jaccard distances (around 0.9), indicating substantial variability in relied-upon features over time. A sharp dip around September 2017 marks a rare period of stability. The Kendall distances show a moderate but steady pattern, reinforcing that both the set and ranking of important features fluctuate significantly across time. In contrast, APIGraph (Figure 7b) shows a gradual downward trend in both measures, suggesting relatively stable feature importance. Overall, SHAP-based analysis highlights LAMDA's volatility compared to APIGraph, underscoring it's value for studying concept drift, continual learning, and model robustness in dynamic malware detection scenarios.

## 5.4 Label Drift Analysis Across Years

**Analysis Setting.** We study how malware sample labels change over time, focusing on cases where a sample initially classified as malicious based on VirusTotal (VT) consensus is later reclassified as benign in a subsequent year, and vice versa. To analyze label drift, we use metadata from both AndroZoo (AZ) and VirusTotal (VT). AndroZoo provides metadata indicating how many VT engines flagged an application as malware at a given point. We first collected this metadata for a year. Then, for the same set of samples, we retrieved updated reports directly from VT in a later year. By comparing the two reports, we track how many samples changed their labels over time.

**Results.** Table 3 summarizes how labels change over time according to VT and AZ. The table reports yearly statistics from 2013 to 2025, including the total number of malware samples (TS) contained in LAMDA for each year. Among these, we report: the number of samples whose labels have changed to benign ($B_C$); the samples where more VT engines now flag them as malware ($D_{Improved}$); the samples where fewer VT engines now flag them as malware ($D_{Weakened}$); and the samples where the total number of VT detections remains unchanged ($D_{Unchanged}$). For example, in 2013, among 44,383 malware samples that were initially detected as malicious, 24 are no longer flagged by any VT engine. Of the remaining 40,436 samples, 439 are now flagged by fewer engines, 3,484 show no change, and the rest are flagged by more engines. In addition, some samples show a significant drop ($DS_{Dropped}$) or increase ($DS_{Increase}$) in detection count, with these columns specifically highlighting drastic change (greater than 50%) in detection count.

## 6 Concept Drift Adaptation

In this section, we evaluate concept drift adaptation (CDA) using three state-of-the-art (SOTA) techniques: Chen et al. (2023), CADE (Yang et al., 2021b), and TRANSCENDENT (Barbero et al., 2022), on the LAMDA dataset to highlight the unique adaptation challenges compared to APIGraph.

We adopt the Chen et al. (2023) active learning framework, which operates on a monthly cycle with labeling budgets of 50, 100, 200, and 400 samples. In each cycle, a subset of test samples is selected for labeling, after which the model is retrained to mitigate performance degradation.

**Results.**    Table 4 summarizes the performance of the evaluated methods. Across most labeling budgets, Chen-AL delivers the strongest results, except at the budget of 400 on LAMDA. Compared to APIGraph, LAMDA introduces substantially more *challenging adaptation scenarios*. While Chen et al. (2023) consistently outperforms CADE and TRANSCENDENT, all methods fail to generalize effectively on LAMDA. For instance, with a labeling budget of 400, Chen et al. (2023) achieves an F1-score of about 92% on APIGraph, but drops sharply to 43% on LAMDA. Similarly, CADE and TRANSCENDENT achieve only 45% and 40% F1-score, respectively. These findings empirically demonstrate that existing SOTA CDA techniques are insufficient when faced with longitudinal and complex distributional shifts such as those captured by LAMDA, highlighting the need for more advanced CDA approaches capable of adapting to long-term drift.

# 7    DISCUSSION AND LIMITATION

LAMDA's temporal structure facilitates evaluation of generalization under distributional shift, while its family diversity allows for studying malware evolution and model adaptation with limited data, including transfer and few-shot learning. Beyond these uses, LAMDA is also designed to facilitate continual learning research for both Domain and Class incremental learning (Rahman et al., 2022; Park et al., 2025), with detailed experimental results provided in Appendix I.

**Limitations.**    While LAMDA provides a solid foundation for studying concept drift in malware analysis, we acknowledge a few limitations. First, it relies exclusively on DREBIN (Arp et al., 2014) features. Other static features such as control flow graphs and clustered API calls are out of scope of this work. Furthermore, we do not perform feature extraction on dynamic behaviors of the APKs which are observable only at runtime. Second, while previous work suggests a 10:90 malware-to-benign ratio, LAMDA attempts to maintain a 50:50 ratio, which may be viewed as downplaying the role of benign software distributions. However, this design emphasizes family diversity and balanced classes to create a challenging benchmark.

**Future Research Direction.**    We envision the following future to tackle the challenges of concept drift adaptation depicted in this work. Firstly, a dynamic learning approach is needed, where the system can recognize the drifted patterns and adjust itself over time. Along with semi-supervised learning, concept adjustment techniques He et al. (2024) and continual learning Park et al. (2025) can enable the system to detect malware more effectively under continuous and evolving drifts. Secondly, multi-modal representation learning may help models handle drifts more effectively. Finally, techniques that can handle *singleton* samples and both intra-class and inter-class distribution shifts would be helpful. In addition, methods that can handle both concept drift and label drifts simultaneously can further improve robustness under the distribution changes observed in LAMDA.

**LAMDA Extension.**    We plan to continue the support for LAMDA and extend this dataset with a focus on multi-modal feature integration. Specifically, we will incorporate not only DREBIN-based static features, but also dynamic sandbox behaviors, control-flow execution characteristics, and enriched threat-intelligence feeds.

# 8    CONCLUSION

In this paper, we present LAMDA, a large-scale, temporally structured Android malware dataset spanning over a decade. It enables analysis of how detection performance evolves with shifts in malware behavior and feature distributions. Our evaluations on supervised learning, feature stability, and explanation analysis demonstrate the impact of these temporal shifts. With broad coverage, diverse families, and static features, LAMDA offers a reproducible benchmark for advancing resilient and adaptive malware detection systems.

ETHICS STATEMENT

This work does not involve human subjects or personally identifiable information. All Android applications were collected from the publicly available AndroZoo repository and contain no user data. To avoid misuse, we provide only extracted feature representations and associated metadata. As we didn't share live malware, the dataset can be made openly available to the research community. We believe the release of LAMDA fully complies with the ICLR Code of Ethics.

REPRODUCIBILITY STATEMENT

We provide comprehensive documentation and scripts to facilitate reproducibility. The dataset is publicly available on Zenodo in both Parquet and NPZ formats, accompanied by detailed metadata, allowing researchers to select the format most suitable for their needs. All code and experimental pipelines used in this work are released through an anonymous GitHub repository, which includes clear execution scripts and step-by-step guidelines to reproduce the reported results. Importantly, our source code is designed to be flexible, enabling researchers to incorporate new samples into the dataset with minimal effort.

In addition, we include detailed documentation of the dataset construction process, metadata, preprocessing steps, and experimental configurations. Together, these resources ensure that the findings of this paper can be independently verified, reproduced, and extended by the research community.

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

## A    DATASET STATISTICS

LAMDA benchmark is constructed from a total of 1,008,381 Android APKs, comprising 369,906 malware samples and 638,475 benign samples. Table 5 summarizes the yearly distribution of both malware and benign APKs. To mitigate class imbalance during training, our initial goal was to collect approximately 50,000 malware and 50,000 benign samples per year; this target could not be met in certain years. Specifically, we were unable to collect sufficient samples for the years 2017, 2021, 2023, 2024, and 2025 due to our labeling criterion—requiring a VirusTotal detection count of 4 or more for malware—and the limited availability of up-to-date samples in the AndroZoo repository (Allix et al., 2016; Alecci et al., 2024). Additional constraints, such as corrupted downloads and decompilation failures, further reduced the effective sample count in those years. Despite these limitations, LAMDA remains the largest Android malware dataset to date in terms of both total sample count and temporal coverage.

Beyond binary labels, LAMDA also includes family-level annotations for malware samples. As shown in Table 6, the dataset spans 1,380 distinct malware families, offering rich diversity for future analysis. Additionally, 150,604 samples are singletons, belonging to families that appear only once in the dataset, representing rare or unique variants. Moreover, 2,985 samples are marked as "unknown", where AVClass2 is unable to confidently assign a family label. Table 7 reports the VirusTotal (VirusTotal, 2025) detection counts for these unknown-labeled samples, offering insight into their potential threat level even in the absence of a family tag.

This comprehensive summary, encompassing both class labels and family-level information, supports a wide range of research directions, including supervised detection, rare variant modeling, family classification, and concept drift analysis across diverse malware behaviors.

## B    FEATURE DESCRIPTION

Built upon static analysis of Android APKs, LAMDA incorporates a broad spectrum of execution-free features based on the features of Drebin (Arp et al., 2014). Table 8 summarizes the key categories of static features used in LAMDA (Arp et al., 2014). These include declared components

Table 5: Year-wise distribution of total, malware, and benign samples.

| Year | Total Samples | Malware Samples | Benign Samples |
|------|---------------|-----------------|----------------|
| 2013 | 86,431 | 44,383 | 42,048 |
| 2014 | 101,183 | 45,756 | 55,427 |
| 2016 | 109,193 | 45,134 | 64,059 |
| 2017 | 99,144 | 21,359 | 77,785 |
| 2018 | 104,292 | 39,350 | 64,942 |
| 2019 | 91,050 | 41,585 | 49,465 |
| 2020 | 102,073 | 46,355 | 55,718 |
| 2021 | 81,155 | 35,627 | 45,528 |
| 2022 | 86,416 | 41,648 | 44,768 |
| 2023 | 54,354 | 7,892 | 46,462 |
| 2024 | 48,427 | 794 | 47,633 |
| 2025 | 44,663 | 23 | 44,640 |
| **Total** | **1,008,381** | **369,906** | **638,475** |

Table 6: Year-wise breakdown of malware family distributions in LAMDA.

| Year | New | Existing | Valid Family | #of Singleton | #of Unknown |
|------|-----|----------|--------------|---------------|-------------|
| 2013 | 213 | 0 | 213 | 1550 | 24 |
| 2014 | 91 | 140 | 231 | 2482 | 345 |
| 2016 | 179 | 196 | 375 | 5861 | 177 |
| 2017 | 88 | 119 | 207 | 9063 | 1108 |
| 2018 | 153 | 220 | 373 | 20579 | 1242 |
| 2019 | 259 | 376 | 635 | 18916 | 22 |
| 2020 | 141 | 447 | 588 | 30644 | 25 |
| 2021 | 43 | 252 | 295 | 30020 | 23 |
| 2022 | 161 | 490 | 651 | 24927 | 4 |
| 2023 | 37 | 187 | 224 | 5922 | 15 |
| 2024 | 14 | 50 | 64 | 626 | 0 |
| 2025 | 1 | 7 | 8 | 14 | 0 |
| **Total** | **1,380** | | | **150,604** | **2,985** |

Table 7: Distribution of unknown malware samples by VirusTotal detection count.

| VT Detection | 4 | 5 | 6 | 7 | 8 | 9 | 10 | 11 | 12 | 13 | 14 | 15 | 18 | 19 | **Total** |
|--------------|---|---|---|---|---|---|----|----|----|----|----|----|----|----|-----------|
| **# of Unknown Sample** | 1643 | 664 | 226 | 153 | 133 | 65 | 68 | 15 | 3 | 4 | 5 | 4 | 1 | 1 | **2,985** |

(e.g., services, activities), permissions (requested and used), intent filters, restricted or suspicious API calls, and embedded network indicators such as hardcoded IPs and URLs.

Each APK is converted into a binary feature vector using a bag-of-tokens representation. Tokens are derived from the presence or absence of the static properties listed in Table 8. Since each application typically uses only a small fraction of the global feature space, the resulting vectors are sparse and high-dimensional. To address this, we apply different `VarianceThreshold` feature selection (Pedregosa et al., 2011), resulting in three dataset variants with different dimensionalities and sizes. Table 9 summarizes these variants. The `Baseline` variant uses a threshold of 0.001 (Rahman et al., 2025) and yields 4,561 binary features. Increasing the threshold to 0.01 results in a smaller, more compressed feature space with 925 features, while lowering it to 0.0001 expands the feature space to over 25,000 features.

Table 8: Static Features and Their Descriptions.

| Feature | Description |
|---|---|
| **Requested permissions** | Permissions declared in the manifest (e.g., CAMERA, BLUETOOTH) indicating intended access to sensitive resources. |
| **Declared activities and services** | Registered components of the application, providing insight into its structural and behavioral composition. |
| **Broadcast receivers** | Components that handle specific system or custom intents (e.g., BOOT_COMPLETED), often linked to persistence or event-driven behavior. |
| **Hardware components** | Device capabilities required by the app (e.g., camera, Bluetooth), implying functional intent. |
| **Intent filters** | Define the types of intents components can respond to; critical for modeling potential entry points. |
| **Used permissions** | Permissions referenced in the smali code, reflecting actual permission usage. |
| **Restricted API calls** | APIs that are protected by system permissions or grant access to sensitive resources. |
| **Suspicious API calls** | APIs heuristically associated with malicious or abnormal behavior. |
| **Embedded IP addresses and URL domains** | Hardcoded network endpoints that may indicate command-and-control (C&C) servers or tracking mechanisms. |

## B.1 LAMDA VARIANTS

For the LAMDA dataset variants, we apply different thresholds using the `VarianceThreshold` (varTh) feature selector. In the baseline configuration (`varTh = 0.001`), we retain 4,561 features with a total in-memory size of 222 MB. For a more relaxed threshold (`varTh = 0.0001`), we preserve 25,460 features, resulting in a memory size of 554 MB. Conversely, applying a stricter threshold (`varTh = 0.01`) yields 915 features with a reduced storage size of 138 MB. These information are summarized in Table 9.

Table 9: Summary of Dataset Variants by Variance Threshold.

| Variant | Threshold | # Metadata | # Binary Features | Size |
|---|---|---|---|---|
| Baseline | 0.001 | 5 | 4561 | 222MB |
| var_thresh_0.0001 | 0.0001 | 5 | 25460 | 554MB |
| var_thresh_0.01 | 0.01 | 5 | 925 | 138MB |

## C ADDITIONAL ANALYSIS OF CONCEPT DRIFT

In this section, we provide in detail analysis of LAMDA visualization using t-SNE and feature space stability analysis using Optimal Transport Dataset Distance (OTDD) (Alvarez-Melis & Fusi, 2020).

### C.1 VISUAL ANALYSIS OF CONCEPT DRIFT

To visualize the structural differences these features capture, we present t-SNE projections comparing LAMDA and API Graph (Dragoi et al., 2022) in Figure 8. LAMDA shows more scattered and diverse malware clusters over time, suggesting richer feature representations and stronger concept drift compared to the relatively compact structure in API Graph. This diversity, driven by the dynamic use of static tokens such as APIs and permissions, highlights the importance of broad and representative feature sets for modeling evolving malware behavior. Figure 13 further validates this hypothesis with varying number of virus total engine detection count.

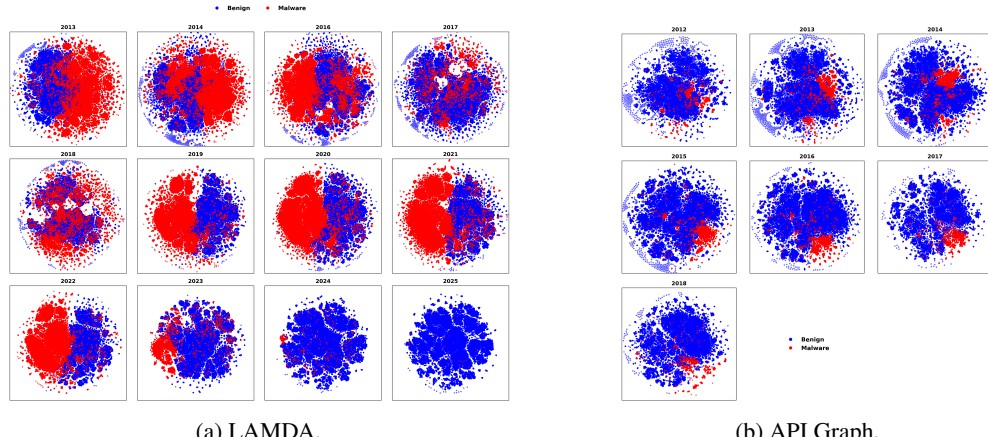

(a) LAMDA.                                    (b) API Graph.

Figure 8: t-SNE projection of LAMDA and API Graph dataset, (a) t-SNE project of LAMDA from 2013 to 2025 (excluding 2015) and (b) t-SNE projection of API Graph from 2012 to 2018.

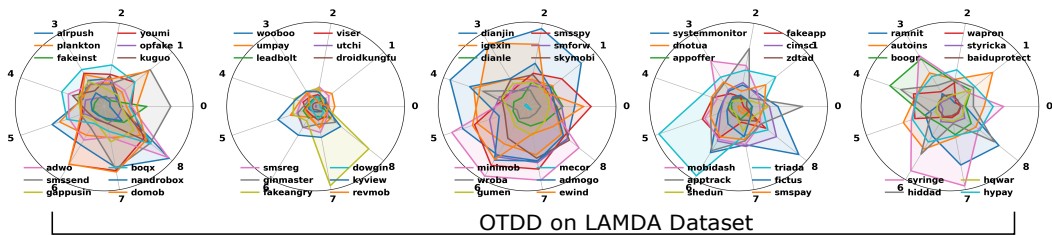

Figure 9: Optimal Transport Distance of 60 common families. Each of the plots shows the area of nine OTDD scores of 10 groups of 10 families in LAMDA.

### C.2 FEATURE SPACE STABILITY ANALYSIS

**Optimal Transport Dataset Distance (OTDD) Analysis.** Figure 9 and 10 illustrates temporal distributional shifts using Optimal Transport Dataset Distance (OTDD) (Alvarez-Melis & Fusi, 2020), a geometric method for quantifying differences between probability distributions. To assess intra-family drift, we partition each malware family in the LAMDA and APIGraph datasets into ten chronological subsets and compute OTDD between consecutive pairs. The results are visualized via radar plots, where each axis represents a subset transition. Compact, regular shapes indicate temporal stability, while larger or irregular shapes signal drift. Comparing the two datasets, the LAMDA radar plots show both regular and irregular patterns indicating temporal shifts of malware families that causes concept drift. Similar behavior is also observed in the APIGraph dataset for the same families.

## D ADDITIONAL EXPERIMENTAL DETAILS

In this section, we summarize the details of the model architectures and the training setup for each method used in our experiments. In addition, we present supplementary results.

### D.1 DETAILS OF THE BASELINE METHODS

**Multi-Layer Perceptron (MLP).** The MLP model used for the experiments is adapted from prior work (Rahman et al., 2022; 2025) and is composed of four fully connected layers with the following sizes: 1024, 512, 256, and 128. Each hidden layer is followed by batch normalization, ReLU activation, and a dropout layer with a dropout rate of 0.5. The final output layer uses a `sigmoid` activation function for binary classification. The model is trained using Adam optimizer with a

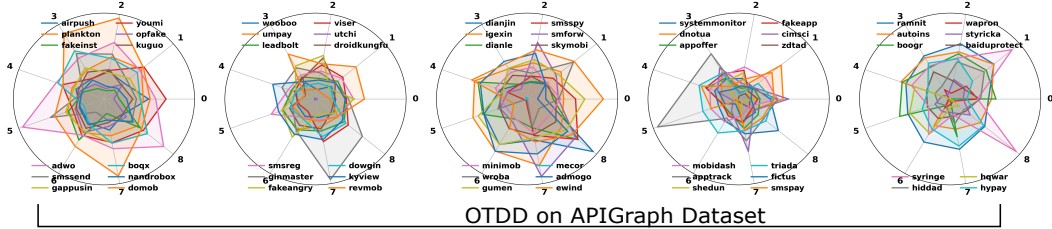

Figure 10: Optimal Transport Distance of 60 common families. Each of the plots shows the area of nine OTDD scores of 10 groups of 10 families in APIGraph.

learning rate of 0.001, and batch size 512, as it stabilizes by this point, avoiding unnecessary GPU time.

**LightGBM.** In addition to MLP, we also use LightGBM (Ke et al., 2017), gradient-boosted decision tree ensemble, for binary classification. LightGBM is trained with up to 5000 estimators and a learning rate of 0.02, with early stopping based on Area Under the Curve (AUC) metric if no improvement is observed for 100 rounds. Each tree is allowed up to 256 leaves to provide high capacity for learning complex patterns. We apply 80% subsampling of both rows and features to mitigate overfitting. We also include $L_1$ and $L_2$ regularization to further penalize methods complexity to prevent overfitting. These hyperparameters are selected based on practices in malware detection benchmarks such as EMBER (Anderson & Roth, 2018) and TESSERACT (Pendlebury et al., 2019).

**XGBoost.** The adapted XGBoost is configured with a tree depth of 12 and a learning rate of 0.05. We use log loss objective for binary classification (Chen & Guestrin, 2016; Anderson & Roth, 2018). The method is trained for up to 3000 boosting rounds and uses the `gpu_hist` tree construction method to accelerate training. The input data is loaded in XGBoost's `DMatrix` format, which is optimized for memory efficiency and fast training. We train the method on the full training data without applying early stopping and evaluate using log loss.

**Support Vector Machine (SVM).** A linear SVM model is implemented using `LinearSVC` and calibrated using `CalibratedClassifierCV` to enable probability outputs. This is essential for downstream evaluation where probabilistic thresholds or ranking-based metrics are used. Following prior work, the method is trained on the full dataset with a maximum of 10,000 iterations (Chen et al., 2023). Post-training, model memory usage is reported using `psutil` to assess resource footprint.

**detectBERT.** DetectBERT is a lightweight transformer-based model designed for malware classification that leverages DexBERT embeddings to learn full app-level representations. It projects the input vector into a hidden space using a fully connected layer, then prepends a learnable `[CLS]` token. The resulting sequence is processed by two stacked transformer layers utilizing Nyström-based self-attention (Xiong et al., 2021), followed by LayerNorm and a classification head applied to the `[CLS]` output. The model also supports alternative aggregation strategies such as averaging or summation over token embeddings.

**ViT.** We utilize a ViT-based model adapts the Vision Transformer (ViT) framework to malware classification using static feature vectors. Each input sample, represented as a flat feature vector, is projected into a hidden-dimensional token via a linear embedding layer. A learnable `[CLS]` token is optionally prepended to the sequence, and positional embeddings are added to all tokens. The resulting token sequence is passed through a stack of seven Transformer encoder blocks, each consisting of LayerNorm, multi-head self-attention, and a two-layer feed-forward network with GELU activation and residual connections. After encoding, the final hidden state of the `[CLS]` token (or the feature token if `[CLS]` is not used) is extracted and passed through a classifier composed of a LayerNorm and a linear output layer. The model outputs logits over malware families and supports fine-grained malware classification or detection.

Table 10: Performance of models across IID, NEAR, and FAR splits for LAMDA on Baseline (`VarianceThreshold = 0.001`).

| Split | Model | Accuracy | Precision | Recall | F1 | ROC AUC | PR AUC | FPR | FNR |
|-------|-------|----------|-----------|--------|-----|---------|--------|-----|-----|
| IID | LightGBM | **97.74 ± 0.35** | 96.74 ± 0.31 | **98.26 ± 0.34** | **97.49 ± 0.17** | **99.55 ± 0.03** | **99.50 ± 0.11** | 2.69 ± 0.48 | **1.74 ± 0.34** |
| | MLP | 97.50 ± 0.44 | 96.91 ± 0.29 | 97.50 ± 0.06 | 97.21 ± 0.12 | 99.48 ± 0.04 | 99.38 ± 0.20 | 2.58 ± 0.85 | 2.50 ± 0.06 |
| | SVM | 95.61 ± 0.61 | 94.78 ± 1.41 | 95.18 ± 0.76 | 94.98 ± 1.07 | 98.89 ± 0.28 | 98.75 ± 0.46 | 4.09 ± 0.55 | 4.82 ± 0.76 |
| | XGBoost | 97.36 ± 0.15 | 96.32 ± 0.70 | 97.80 ± 0.43 | 97.05 ± 0.14 | 99.15 ± 0.16 | 97.68 ± 1.16 | 2.96 ± 0.01 | 2.20 ± 0.43 |
| | detectBERT | 95.83±0.15 | 95.04±1.12 | 95.51±0.60 | 95.27±0.86 | 98.93±0.27 | 98.62±0.79 | 3.96±0.14 | 4.49±0.60 |
| | ViT | 95.57±0.80 | 94.14±2.42 | 95.83±0.73 | 94.97±1.59 | 98.91±0.37 | 98.61±0.85 | 4.63±0.81 | 4.17±0.73 |
| NEAR | LightGBM | **85.83 ± 3.96** | **90.36 ± 5.21** | **49.49 ± 30.82** | **59.48 ± 28.20** | 74.05 ± 23.76 | **70.18 ± 27.10** | **1.85 ± 0.95** | **50.51 ± 30.82** |
| | MLP | 83.90 ± 3.75 | 78.12 ± 13.98 | 48.05 ± 30.42 | 56.57 ± 28.41 | **82.71 ± 11.19** | 67.94 ± 24.59 | 3.98 ± 1.19 | 51.95 ± 30.42 |
| | SVM | 82.08 ± 3.11 | 72.56 ± 18.38 | 44.38 ± 28.88 | 52.91 ± 28.40 | 75.18 ± 17.98 | 62.53 ± 29.82 | 4.71 ± 0.97 | 55.62 ± 28.88 |
| | XGBoost | 84.59 ± 3.75 | 86.18 ± 9.02 | 46.16 ± 30.94 | 55.84 ± 29.73 | 77.75 ± 16.85 | 68.14 ± 26.55 | 2.14 ± 0.86 | 53.84 ± 30.94 |
| | detectBERT | 84.21±4.81 | 78.61±19.05 | 50.08±37.55 | 59.11±34.70 | 73.86±29.73 | 66.95±37.60 | 4.05±0.84 | 49.92±37.55 |
| | ViT | 83.66 ± 5.36 | 73.73 ± 22.28 | 51.32 ± 39.01 | 58.77 ± 35.48 | 71.47 ± 32.31 | 63.97 ± 40.01 | 5.63 ± 1.28 | 48.68 ± 39.01 |
| FAR | LightGBM | 83.94 ± 10.61 | 74.65 ± 34.66 | 35.90 ± 22.97 | 47.24 ± 27.33 | 78.04 ± 20.83 | 63.45 ± 35.80 | 1.30 ± 0.95 | 64.10 ± 22.97 |
| | MLP | 83.45 ± 10.74 | **76.12 ± 33.39** | 35.60 ± 20.63 | **47.59 ± 25.30** | **84.04 ± 11.23** | 66.16 ± 34.34 | **1.14 ± 0.71** | **64.40 ± 20.63** |
| | SVM | 80.99 ± 11.98 | 72.89 ± 35.60 | 30.07 ± 16.85 | 41.86 ± 22.55 | 79.07 ± 15.06 | 62.27 ± 34.09 | 1.27 ± 0.76 | 69.93 ± 16.85 |
| | XGBoost | 82.03 ± 11.07 | 70.00 ± 37.26 | 31.89 ± 20.43 | 42.75 ± 25.86 | 76.85 ± 16.49 | 60.33 ± 35.88 | 1.69 ± 0.57 | 68.11 ± 20.43 |
| | detectBERT | 81.79±11.09 | 66.49±38.11 | 35.43±20.54 | 45.01±26.84 | 76.37±23.07 | 60.56±36.63 | 3.15±1.09 | 64.57±20.54 |
| | ViT | 81.98 ± 9.44 | 63.56 ± 38.55 | 39.77 ± 22.66 | 47.03 ± 29.55 | 78.53 ± 20.17 | 58.77 ± 37.82 | 4.77 ± 1.20 | 60.23 ± 22.66 |

All models are trained on three different LAMDA variants with `VarianceThreshold` (VarTh) $\in \{0.01, 0.001, 0.0001\}$ where VarTh = 0.001 is the baseline. No task-specific tuning or dataset-specific hyperparameter adjustments are performed to ensure fair comparisons across splits and datasets.

## D.2 BASELINE PERFORMANCE

We compare LAMDA baseline with API Graph (Zhang et al., 2020) dataset and provide a comprehensive results on four methods discussed above using AnoShift-style (Dragoi et al., 2022) splits. A subset of Table 10 and Table 11 are explained in the main body of the paper. We present the results with more performance metrics.

Across both NEAR and FAR splits, LAMDA consistently exhibits lower scores across all performance metrics compared to API Graph, and notably higher false negative rates (FNR). These trends clearly indicate that LAMDA captures a significantly higher degree of concept drift. Furthermore, the standard deviation across metrics is substantially higher in LAMDA, especially for drifted years, underscoring the dataset's temporal instability in detection performance—validating the presence of concept drift.

Table 11: Performance of models on across IID, NEAR, and FAR splits for API Graph.

| Split | Model | Accuracy | Precision | Recall | F1 | ROC AUC | PR AUC | FPR | FNR |
|-------|-------|----------|-----------|--------|-----|---------|--------|-----|-----|
| IID | LightGBM | 97.02 ± 0.00 | **95.78 ± 0.00** | 73.44 ± 0.00 | 83.14 ± 0.00 | **98.93 ± 0.00** | **94.92 ± 0.00** | **0.36 ± 0.00** | 26.56 ± 0.00 |
| | MLP | 97.35 ± 0.20 | 94.84 ± 1.49 | 77.79 ± 2.85 | 85.43 ± 1.35 | 94.62 ± 0.92 | 89.33 ± 1.75 | 0.47 ± 0.16 | 22.21 ± 2.85 |
| | SVM | 96.64 ± 0.00 | 94.12 ± 0.00 | 70.85 ± 0.00 | 80.84 ± 0.00 | 97.27 ± 0.00 | 90.90 ± 0.00 | 0.49 ± 0.00 | 29.15 ± 0.00 |
| | XGBoost | 96.30 ± 0.00 | 91.57 ± 0.00 | 69.37 ± 0.00 | 78.94 ± 0.00 | 95.93 ± 0.00 | 89.08 ± 0.00 | 0.71 ± 0.00 | 30.63 ± 0.00 |
| | detectBERT | 97.01 ± 0.00 | 94.59 ± 0.00 | 74.40 ± 0.00 | 83.05 ± 0.00 | 98.82 ± 0.00 | 94.19 ± 0.00 | 0.48 ± 0.00 | 25.60 ± 0.00 |
| | ViT | 97.49±0.00 | 91.85±0.00 | 82.18±0.00 | 86.64±0.00 | 98.65±0.00 | 93.73±0.00 | 0.81±0.00 | 17.82±0.00 |
| NEAR | LightGBM | **94.84 ± 0.00** | **94.07 ± 0.00** | 51.33 ± 0.00 | 66.42 ± 0.00 | **96.28 ± 0.00** | **83.45 ± 0.00** | **0.36 ± 0.00** | **48.67 ± 0.00** |
| | MLP | 94.66 ± 0.34 | 88.22 ± 3.83 | 53.67 ± 4.84 | 66.55 ± 3.19 | 85.97 ± 1.25 | 72.24 ± 2.46 | 0.81 ± 0.37 | 46.33 ± 4.84 |
| | SVM | 94.01 ± 0.00 | 81.70 ± 0.00 | 51.18 ± 0.00 | 62.93 ± 0.00 | 90.29 ± 0.00 | 70.84 ± 0.00 | 1.26 ± 0.00 | 48.82 ± 0.00 |
| | XGBoost | 92.76 ± 0.00 | 77.60 ± 0.00 | 38.11 ± 0.00 | 51.12 ± 0.00 | 92.13 ± 0.00 | 65.74 ± 0.00 | 1.21 ± 0.00 | 61.89 ± 0.00 |
| | detectBERT | 95.24 ± 0.29 | 90.54 ± 1.04 | 56.87 ± 0.71 | 69.08 ± 0.69 | 94.33 ± 0.45 | 82.54 ± 1.20 | 0.59 ± 0.15 | 43.13 ± 0.71 |
| | ViT | 94.73±0.00 | 76.15±0.00 | 68.77±0.00 | 72.15±0.00 | 93.08±0.00 | 78.26±0.00 | 2.41±0.00 | 31.23±0.00 |
| FAR | LightGBM | **95.31 ± 0.61** | **87.48 ± 2.48** | 57.41 ± 6.57 | 69.07 ± 4.73 | 96.11 ± 0.40 | 80.94 ± 1.17 | 0.84 ± 0.24 | 42.59 ± 6.57 |
| | MLP | 94.50 ± 0.84 | 82.31 ± 6.28 | 51.14 ± 7.90 | 62.80 ± 6.68 | 90.02 ± 2.91 | 71.38 ± 5.89 | 1.12 ± 0.43 | 48.86 ± 7.90 |
| | SVM | 94.33 ± 0.60 | 77.38 ± 0.33 | 54.34 ± 7.27 | 63.56 ± 5.08 | 92.73 ± 0.92 | 71.46 ± 2.43 | 1.60 ± 0.20 | 45.66 ± 7.27 |
| | XGBoost | 93.70 ± 0.35 | 75.99 ± 2.62 | 46.04 ± 1.92 | 57.29 ± 1.45 | 87.78 ± 1.90 | 66.49 ± 4.80 | 1.49 ± 0.28 | 53.96 ± 1.92 |
| | detectBERT | 95.85 ± 0.53 | 86.50 ± 1.92 | 64.33 ± 5.55 | 73.23 ± 3.68 | 96.74 ± 0.17 | 83.15 ± 1.44 | 0.99 ± 0.15 | 35.67 ± 5.55 |
| | ViT | 94.01±0.71 | 65.50±1.51 | 72.29±6.81 | 68.47±3.94 | 95.28±0.19 | 73.69±3.83 | 3.81±0.14 | 27.71±6.81 |

## D.3 LAMDA VARIANTS AND DRIFT SENSITIVITY

LAMDA offers flexibility to researchers for Android malware analysis by supporting different feature selection variants. In this section, we evaluate two additional variants of LAMDA. As shown in Table 12 and Table 13, we report detailed performance results for the four methods and configurations used in the primary analysis of concept drift with `VarianceThreshold` of 0.001. The

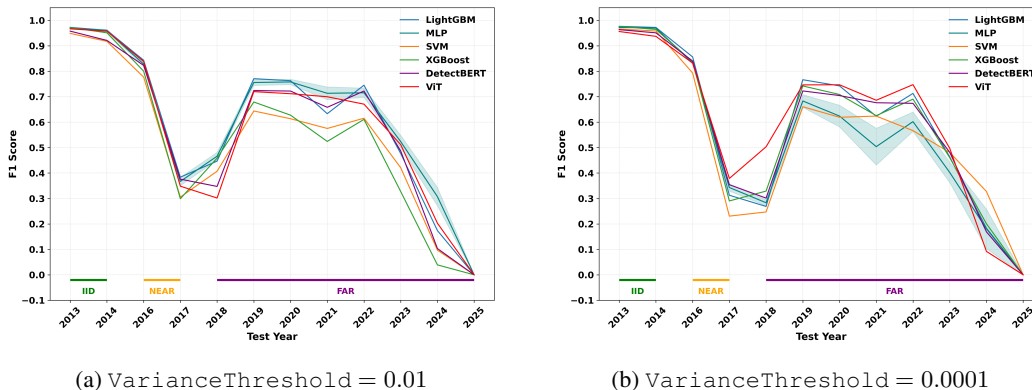

(a) `VarianceThreshold = 0.01`  (b) `VarianceThreshold = 0.0001`

Figure 11: F1-scores on different models on based on AnoShift-style split on LAMDA. (a) for `VarianceThreshold` 0.01 and (b) for `VarianceThreshold` 0.0001.

Table 12: Performance of models across IID, NEAR, and FAR splits for LAMDA variant of `VarianceThreshold` (0.01).

| Split | Model | Accuracy | Precision | Recall | F1 | ROC AUC | PR AUC | FPR | FNR |
|---|---|---|---|---|---|---|---|---|---|
| IID | LightGBM | **97.08 ± 0.21** | **96.40 ± 0.22** | **96.95 ± 1.02** | **96.67 ± 0.61** | **99.42 ± 0.00** | **99.33 ± 0.13** | 2.94 ± 0.43 | 3.05 ± 1.02 |
| | MLP | 96.89 ± 0.32 | 96.11 ± 0.37 | 96.84 ± 1.26 | 96.47 ± 0.67 | 99.29 ± 0.08 | 99.15 ± 0.18 | 3.18 ± 0.54 | 3.16 ± 1.26 |
| | SVM | 94.13 ± 0.87 | 92.59 ± 2.43 | 94.10 ± 0.84 | 93.33 ± 1.65 | 97.86 ± 0.54 | 97.34 ± 1.15 | 5.83 ± 0.90 | 5.90 ± 0.84 |
| | XGBoost | 96.65 ± 0.60 | 95.59 ± 1.37 | 96.74 ± 0.80 | 96.16 ± 1.09 | 99.01 ± 0.26 | 98.00 ± 0.36 | 3.45 ± 0.43 | 3.26 ± 0.80 |
| | detectBERT | 94.75 ± 1.53 | 93.58 ± 3.04 | 94.34 ± 2.10 | 93.96 ± 2.58 | 98.47 ± 0.51 | 98.13 ± 1.02 | 4.98 ± 1.15 | 5.66 ± 2.10 |
| | ViT | 96.66 ± 0.07 | 96.02 ± 0.37 | 96.42 ± 0.93 | 96.22 ± 0.65 | 99.07 ± 0.38 | 98.80 ± 0.82 | 3.24 ± 0.59 | 3.58 ± 0.93 |
| NEAR | LightGBM | **84.17 ± 3.77** | 75.03 ± 15.03 | **54.00 ± 27.34** | **61.38 ± 24.22** | 73.58 ± 22.25 | 66.97 ± 27.37 | 5.86 ± 0.94 | **46.00 ± 27.34** |
| | MLP | 83.39 ± 3.52 | 73.64 ± 16.21 | 52.23 ± 26.78 | 59.77 ± 24.33 | **79.17 ± 14.79** | 66.60 ± 24.33 | 6.09 ± 1.22 | 47.77 ± 26.78 |
| | SVM | 75.73 ± 6.47 | 54.52 ± 25.76 | 53.78 ± 23.97 | 54.13 ± 24.86 | 74.88 ± 14.32 | 58.72 ± 25.70 | 17.09 ± 2.87 | 46.22 ± 23.97 |
| | XGBoost | 81.63 ± 3.39 | 68.78 ± 20.03 | 47.48 ± 27.52 | 54.98 ± 26.43 | 77.32 ± 14.71 | 62.60 ± 26.89 | 6.57 ± 0.68 | 52.52 ± 27.52 |
| | detectBERT | 82.23 ± 5.19 | 67.95 ± 24.83 | 54.66 ± 34.89 | 59.95 ± 31.61 | 71.17 ± 30.38 | 63.83 ± 35.68 | 8.79 ± 0.97 | 45.34 ± 34.89 |
| | ViT | 83.91 ± 5.34 | **75.41 ± 22.66** | 51.05 ± 37.34 | 59.35 ± 34.67 | 72.45 ± 31.57 | **64.86 ± 39.94** | 4.87 ± 0.34 | 48.95 ± 37.34 |
| FAR | LightGBM | 84.68 ± 9.54 | 71.15 ± 35.65 | 39.50 ± 23.10 | 50.13 ± 27.27 | 75.80 ± 21.64 | 61.66 ± 35.89 | 2.34 ± 1.72 | 60.50 ± 23.10 |
| | MLP | **85.15 ± 9.23** | **76.93 ± 33.45** | **41.38 ± 21.41** | **53.00 ± 25.53** | **87.00 ± 10.48** | **68.52 ± 33.91** | 1.80 ± 1.61 | **58.62 ± 21.41** |
| | SVM | 77.50 ± 10.05 | 55.00 ± 32.97 | 36.40 ± 16.42 | 42.16 ± 23.50 | 76.19 ± 12.22 | 52.48 ± 31.86 | 9.28 ± 3.56 | 63.60 ± 16.42 |
| | XGBoost | 76.73 ± 6.52 | 56.84 ± 36.66 | 34.05 ± 17.19 | 40.84 ± 25.02 | 69.95 ± 15.23 | 51.85 ± 34.08 | 8.71 ± 3.04 | 65.95 ± 17.19 |
| | detectBERT | 81.52 ± 9.77 | 60.48 ± 37.30 | 40.21 ± 22.98 | 47.06 ± 29.24 | 76.28 ± 21.99 | 57.68 ± 36.07 | 6.29 ± 2.14 | 59.79 ± 22.98 |
| | ViT | 82.84 ± 10.71 | 67.79 ± 38.23 | 38.47 ± 21.47 | 47.74 ± 27.67 | 77.55 ± 24.71 | 61.82 ± 37.79 | 2.85 ± 1.43 | 61.53 ± 21.47 |

variant using a threshold of 0.01 exhibits a relatively higher average F1-score across NEAR and FAR splits compared to that of the Baseline (`0.001`) and `var_thresh_0.0001` variants. Figure 11 shows this trend as well, highlighting the comparative performance of the LAMDA variants.

To further understand how these feature selection variants influence drift sensitivity, we focus on the NEAR split with the SVM. For false positive rate (FPR), both the baseline and `varTh=0.0001` maintain relatively low values ($\sim$4.07 $\pm$ 0.18), whereas `varTh=0.01` shows a sharp increase to 17.09 $\pm$ 2.87. This suggests that reducing the feature set too aggressively may misclassify benign as malware. Conversely, the false negative rate (FNR) slightly improves under `varTh=0.01`, decreasing from $\sim$55.62 $\pm$ 28.88 and $\sim$57.72 $\pm$ 28.87, baseline and `varTh=0.0001`, respectively, to 46.22 $\pm$ 23.97. This indicates that even with fewer features, the model may still capture certain generalizable malware traits, improving detection of some malicious samples.

However, for the LightGBM model, this change is accompanied by a drop in precision. While in NEAR region both the baseline and `varTh=0.0001` variants maintain high precision scores (around 90.36 $\pm$ 5.21), the `varTh=0.01` variant yields a lower precision of 75.03 $\pm$ 15.03. This reflects a shift in the methods decision behavior under more aggressive feature selection, emphasizing the importance of balancing dimensionality reduction with predictive consistency.

In summary, while `varTh=0.01` may occasionally help with generalization under drift, it also amplifies misclassification of benign apps and reduces predictive stability. The baseline and `varTh=0.0001` offer more shift in data distribution.

Table 13: Performance of models across IID, NEAR, and FAR splits for LAMDA variant of `VarianceThreshold` (0.0001).

| Split | Model | Accuracy | Precision | Recall | F1 | ROC AUC | PR AUC | FPR | FNR |
|---|---|---|---|---|---|---|---|---|---|
| IID | LightGBM | **97.73 ± 0.04** | **96.80 ± 0.33** | **98.11 ± 0.12** | **97.45 ± 0.23** | **99.58 ± 0.00** | **99.54 ± 0.09** | **2.61 ± 0.25** | **1.89 ± 0.12** |
| | MLP | 97.42 ± 0.21 | 96.44 ± 0.35 | 97.80 ± 0.23 | 97.12 ± 0.15 | 99.31 ± 0.06 | 99.24 ± 0.17 | 2.92 ± 0.41 | 2.20 ± 0.23 |
| | SVM | 96.68 ± 0.02 | 96.68 ± 0.08 | 95.77 ± 0.72 | 96.22 ± 0.40 | 99.13 ± 0.20 | 99.13 ± 0.29 | 2.69 ± 0.48 | 4.23 ± 0.72 |
| | XGBoost | 97.39 ± 0.26 | 96.27 ± 0.69 | 97.83 ± 0.53 | 97.04 ± 0.61 | 99.35 ± 0.08 | 98.92 ± 0.10 | 3.01 ± 0.04 | 2.17 ± 0.53 |
| | detectBERT | 96.24 ± 0.23 | 95.33 ± 0.90 | 96.15 ± 0.87 | 95.74 ± 0.89 | 98.92 ± 0.40 | 98.61 ± 0.93 | 3.76 ± 0.28 | 3.85 ± 0.87 |
| | ViT | 95.27 ± 0.50 | 93.32 ± 1.29 | 96.08 ± 1.41 | 94.68 ± 1.35 | 98.76 ± 0.34 | 98.56 ± 0.69 | 5.49 ± 0.41 | 3.92 ± 1.41 |
| NEAR | LightGBM | **85.55 ± 3.91** | **90.24 ± 5.75** | 48.34 ± 30.74 | 58.46 ± 28.66 | 74.58 ± 23.25 | **70.47 ± 26.85** | **1.70 ± 0.80** | 51.66 ± 30.74 |
| | MLP | 84.22 ± 3.38 | 80.23 ± 13.34 | 49.38 ± 27.83 | 58.79 ± 25.81 | 81.65 ± 13.28 | 69.02 ± 25.24 | 3.71 ± 0.95 | 50.62 ± 27.83 |
| | SVM | 81.79 ± 3.36 | 71.61 ± 21.55 | 42.28 ± 28.87 | 51.19 ± 29.64 | 76.15 ± 16.45 | 63.54 ± 29.13 | 4.07 ± 0.18 | 57.72 ± 28.87 |
| | XGBoost | 84.58 ± 3.41 | 86.82 ± 6.81 | 46.86 ± 30.72 | 56.40 ± 28.84 | 77.29 ± 18.06 | 69.61 ± 25.51 | 2.52 ± 1.41 | 53.14 ± 30.72 |
| | detectBERT | 84.30 ± 4.99 | 77.92 ± 19.62 | 50.98 ± 37.46 | 59.73 ± 34.36 | 74.34 ± 28.90 | 67.51 ± 36.87 | 4.33 ± 0.78 | 49.02 ± 37.46 |
| | ViT | 81.49 ± 6.49 | 63.97 ± 26.76 | 58.13 ± 36.00 | 60.56 ± 32.07 | 69.96 ± 31.48 | 59.47 ± 40.55 | 11.55 ± 0.86 | 41.87 ± 36.00 |
| FAR | LightGBM | **83.96 ± 10.66** | **75.34 ± 34.34** | **35.62 ± 23.11** | **47.02 ± 27.47** | 77.98 ± 21.70 | **64.21 ± 35.59** | **1.18 ± 0.85** | 64.38 ± 23.11 |
| | MLP | 80.88 ± 11.82 | 71.93 ± 36.09 | 29.64 ± 17.51 | 40.99 ± 23.11 | 82.52 ± 12.80 | 63.53 ± 35.27 | 1.47 ± 1.11 | 70.36 ± 17.51 |
| | SVM | 81.23 ± 11.94 | 73.48 ± 34.34 | 32.40 ± 16.44 | 44.08 ± 21.95 | 76.94 ± 16.51 | 63.06 ± 32.51 | 1.31 ± 0.86 | 67.60 ± 16.44 |
| | XGBoost | 83.47 ± 10.41 | 73.53 ± 35.83 | 35.43 ± 20.83 | 46.97 ± 25.75 | 77.49 ± 19.29 | 63.56 ± 34.89 | 1.32 ± 0.67 | 64.57 ± 20.83 |
| | detectBERT | 82.54 ± 10.53 | 67.37 ± 38.73 | 37.34 ± 21.46 | 46.62 ± 27.85 | 77.47 ± 22.32 | 61.65 ± 37.23 | 2.92 ± 1.21 | 62.66 ± 21.46 |
| | ViT | 82.03 ± 7.65 | 59.73 ± 37.03 | 45.42 ± 23.33 | 50.25 ± 30.10 | 76.63 ± 18.94 | 58.53 ± 36.04 | 8.09 ± 2.42 | 54.58 ± 23.33 |

# E  BEHIND THE SCENES: PRACTICAL CHALLENGES IN LAMDA CREATION

## E.1  ADMINISTRATIVE CHALLENGES

Downloading large volumes of real-world malware presents significant cybersecurity risks within any institutional environment. During our data collection process, the downloading and unpacking of live malware samples triggered internal threat detection systems, as automated security tools flagged these activities as potential breaches. To mitigate these risks, we implemented strict containment policies, such as disabling execution permissions. Additionally, we worked closely with the university's cybersecurity team to obtain the necessary approvals and ensure compliance with all relevant security policies. We maintained continuous communication with them throughout the process to ensure proper coordination and promptly address any emerging issues.

## E.2  TECHNICAL CHALLENGES

We faced several technical constraints during the sample collection and processing pipeline. The AndroZoo platform imposes strict download rate limits, allowing only 40 concurrent downloads per user. As a result, we had to be extremely cautious to avoid violating their terms and conditions. Unfortunately, accidental oversights on our part led to temporary request blocks which disrupted the collection process. Similarly, the VirusTotal API has strict rate limits, which can significantly slow down the retrieval of metadata. Additionally, a considerable number of APKs failed to decompile successfully using Apktool. These failures were often due to obfuscation, corrupted files, or nonstandard packaging formats. So, we had to perform multiple rounds of sampling to reach our target number of usable samples.

# F  EFFECT OF LABEL NOISE IN TRAINING DATA

Label noise in Android malware family classification, particularly when using the Drebin (Arp et al., 2014) feature set can significantly impact model performance due to ambiguous and overlapping feature representations. As demonstrated in recent work (Oyen et al., 2022), the robustness of classification models depends not only on the amount of label noise but also on its distribution within the feature space. Specifically, feature-dependent label noise, where the probability of a label flip is contingent on the position of a sample in feature space, can cause a substantial drop in accuracy, even at low noise levels.

This is especially relevant for Drebin features, where different malware families may share static features like permissions ($x_1 =$ INTERNET, $x_2 =$ SEND_SMS), API calls ($x_3 =$ getDeviceId), and hardware access ($x_4 =$ READ_PHONE_STATE). Samples with minimal or ambiguous patterns (e.g., $x_5 =$ ACCESS_NETWORK_STATE and $x_6 =$ RECEIVE_BOOT_COMPLETED) are likely to fall near decision boundaries, increasing the risk of mislabeling. Such feature-dependent noise is more

Table 14: Effect of thresholding on sample counts and relative percentage change (w.r.t. threshold 4).

| Threshold | Benign Samples | Malware Samples | % Change (Benign) | % Change (Malware) |
|---|---|---|---|---|
| 1 | 638,475 | 369,906 | 0.0% | 0.0% |
| 2 | 638,475 | 369,906 | 0.0% | 0.0% |
| 3 | 638,475 | 369,906 | 0.0% | 0.0% |
| 4 | 638,475 | 369,906 | 0.0% | 0.0% |
| 5 | 638,475 | 324,927 | 0.0% | ↓ 12.16% |
| 6 | 638,475 | 281,824 | 0.0% | ↓ 23.81% |
| 7 | 638,475 | 241,690 | 0.0% | ↓ 34.66% |
| 8 | 638,475 | 206,644 | 0.0% | ↓ 44.14% |
| 9 | 638,475 | 177,707 | 0.0% | ↓ 51.96% |
| 10 | 638,475 | 155,376 | 0.0% | ↓ 58.00% |
| 11 | 638,475 | 138,041 | 0.0% | ↓ 62.68% |
| 12 | 638,475 | 123,783 | 0.0% | ↓ 66.53% |
| 13 | 638,475 | 111,350 | 0.0% | ↓ 69.89% |

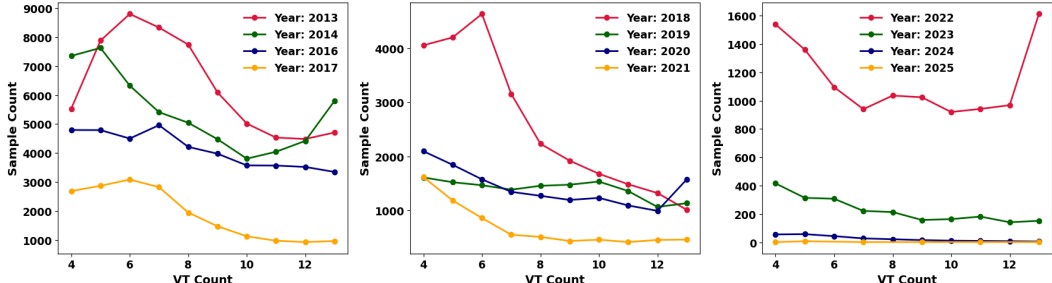

Figure 12: Sample Count Year-wise for Each VT Detection Threshold.

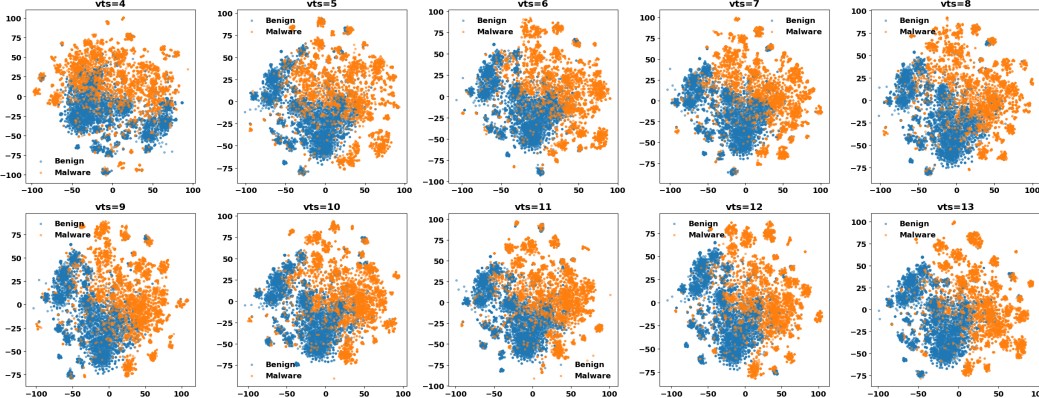

Figure 13: t-SNE visualization of benign and malware samples at varying Virus Total (VT) detection threshold.

detrimental than uniform or class-dependent noise and warrants careful consideration in malware classification tasks.

Table 14 shows the effect of increasing the VirusTotal (VT) detection (VirusTotal, 2025) threshold on malware labeling in the LAMDA dataset. According to previous studies (Pendlebury et al., 2019; Rahman et al., 2022; Yang et al., 2021b), a sample is considered benign if `vt_detection` = 0, and labeled as malware if `vt_detection` ≥ 4. As seen from the Benign sample column in the Table 14, the number of benign samples remains unchanged across all thresholds, since the benign definition is fixed and independent of the malware thresholding rule. However, the number of malware samples decreases significantly as the threshold increases from 4 to 13. For instance, at a

threshold of 10, the number of malware samples drops by 58% compared to the baseline at threshold 4. This trend continues, reaching a 69.89% reduction at threshold 13. These results demonstrate that requiring stronger agreement among antivirus engines (i.e., a higher threshold) leads to more conservative malware labeling, effectively excluding a substantial portion of potentially malicious samples. While this may improve the confidence in the labeled malware, it also drastically reduces dataset coverage. Therefore, the choice of VT threshold directly impacts the balance between label precision and data availability, and threshold 4 provides a practical trade-off commonly adopted in existing literature (Xu et al., 2019; Pendlebury et al., 2019; Park et al., 2025).

Figure 12 illustrates the distribution of malware sample counts across different VirusTotal (VT) threshold (VirusTotal, 2025) values for each year from 2013 to 2025 (except 2015). The VT count, plotted on the $x$-axis, represents the number of antivirus (AV) engines that flagged a sample as malicious, serving as a metric for detection consensus or confidence. A consistent trend is observed across all years — as the VT threshold increases, the number of flagged samples decreases. This suggests that only a small fraction of malware samples achieve strong consensus among AV engines, while the majority are detected by relatively few engines. The sample count is highest between VT counts of 5 to 7, especially in earlier years such as 2013–2017, indicating a moderate level of agreement in those periods. In contrast, from 2022 onward, the overall volume of detected samples decreases sharply, and the detections are largely concentrated in the lower VT ranges, which may reflect advancements in malware evasion techniques or shifts in detection criteria. These observations justify the use of a VT threshold. Using a higher threshold (e.g., $\geq 10$) may lead to overly conservative labeling with potential false negatives, while lower thresholds increase coverage but may introduce noise. Thus, this temporal analysis provides critical insight into threshold selection and highlights the evolving nature of malware detection over time.

Figure 13 shows the t-SNE projections across varying VirusTotal (VT) detection (VirusTotal, 2025) thresholds. At lower thresholds (e.g., VT $\geq 4$ to 6), there is significant overlap between the malware and benign clusters, indicating that many samples labeled as malware may exhibit similar characteristics to benign samples. This suggests that lower thresholds capture a broader range of potentially ambiguous or borderline malicious behaviors. As the VT detection count increases (e.g., VT $\geq 10$), the overlap diminishes, and malware samples become more distinct and spatially separated from benign samples in the embedded space. This indicates that higher-threshold malware samples possess more distinguishable feature representations, likely reflecting stronger and more consistent malicious behaviors detected by a greater number of antivirus engines. Furthermore, the density of malware samples decreases as the threshold rises, aligning with the observed reduction in malware counts from the dataset.

Table 15: F1 scores of the baseline malware detectors with varying VT thresholds.

| Split | Model | VT=4 | VT=5 | VT=6 | VT=7 | VT=8 | VT=9 | VT=10 | VT=11 | VT=12 | VT=13 |
|-------|-------|------|------|------|------|------|------|-------|-------|-------|-------|
| IID | LightGBM | 0.8515 | 0.8566 | 0.8148 | 0.8397 | 0.8172 | 0.8511 | 0.8574 | 0.8364 | 0.6889 | 0.8385 |
| | MLP | 0.8364 | 0.8173 | 0.8166 | 0.8319 | 0.8393 | 0.8232 | 0.8102 | 0.8628 | 0.7536 | 0.8623 |
| | SVM | 0.7898 | 0.7954 | 0.7605 | 0.7543 | 0.8153 | 0.7492 | 0.7598 | 0.7353 | 0.6559 | 0.7462 |
| | XGBoost | 0.8456 | 0.8028 | 0.8298 | 0.8255 | 0.8119 | 0.8370 | 0.8273 | 0.7844 | 0.7324 | 0.7538 |
| NEAR | LightGBM | 0.2423 | 0.2008 | 0.2308 | 0.1994 | 0.2470 | 0.1962 | 0.1956 | 0.2178 | 0.1639 | 0.2133 |
| | MLP | 0.1971 | 0.1894 | 0.2191 | 0.2201 | 0.2223 | 0.2275 | 0.2249 | 0.1771 | 0.1867 | 0.2098 |
| | SVM | 0.1330 | 0.1512 | 0.1569 | 0.1763 | 0.1971 | 0.2712 | 0.1386 | 0.1381 | 0.1261 | 0.1865 |
| | XGBoost | 0.2221 | 0.2064 | 0.2703 | 0.2788 | 0.2761 | 0.1937 | 0.2788 | 0.2361 | 0.2563 | 0.2241 |
| FAR | LightGBM | 0.1284 | 0.1477 | 0.1559 | 0.1039 | 0.0852 | 0.1562 | 0.1551 | 0.0935 | 0.0511 | 0.2306 |
| | MLP | 0.1284 | 0.2716 | 0.0918 | 0.0721 | 0.1362 | 0.1873 | 0.1954 | 0.0841 | 0.0912 | 0.0947 |
| | SVM | 0.2393 | 0.2272 | 0.1503 | 0.0655 | 0.0938 | 0.1104 | 0.1322 | 0.0540 | 0.0508 | 0.0706 |
| | XGBoost | 0.1560 | 0.3513 | 0.2926 | 0.2613 | 0.1971 | 0.1787 | 0.2133 | 0.1067 | 0.1385 | 0.1452 |

To assess the impact of label noise on malware detection, we conduct a set of experiment varying the VirusTotal (VT) labeling Threshold. We first create a set of LAMDA datasets, where in each dataset contain *all benign samples* and *only malware samples with a specific VT labeling count*. For example, for the first dataset, we keep all benign samples and those malware samples that were flagged exactly by 4 antivirus engines (VT=4). In the next dataset, we include malware samples with VT=5, and so on, up to VT=13. This resulted in creating ten separate LAMDA dataset variants, each reflecting a different level of confidence in the AV engines.

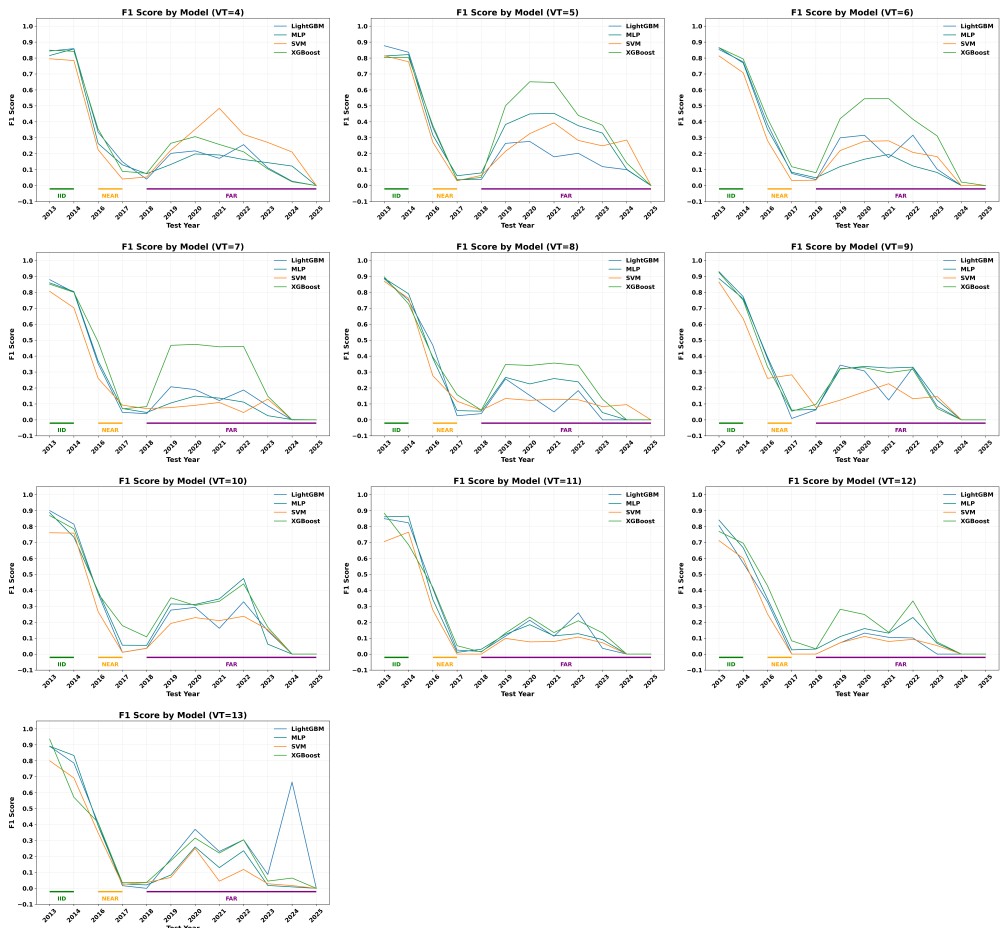

Figure 14: Combined F1 score plots for VT thresholds 4 to 13.

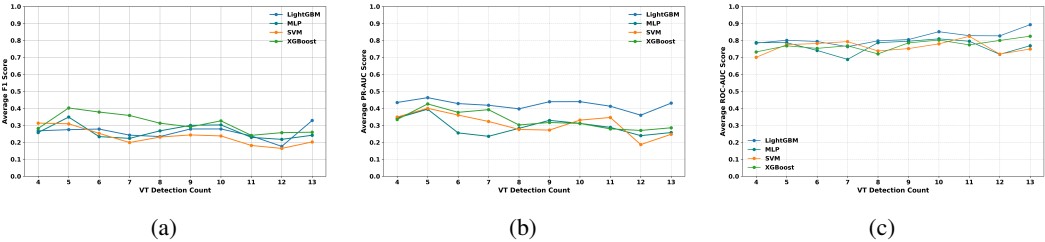

(a)        (b)        (c)

Figure 15: Comparison of model performance across VirusTotal detection thresholds for (a) F1-score, (b) PR-AUC, and (c) ROC-AUC.

Next, We evaluate standard malware detectors (LightGBM, MLP, SVM, and XGBoost) on these datasets using the *AnoShift*-style splits, which simulate temporal concept drift by training on IID split and tested on NEAR and FAR splits. Each malware detector's performance is evaluated using F1-score metric.

Table 15 presents performance details of the baseline malware detectors using F1-scores metric with varying VT count. We made the following observations. LightGBM with VT=4, we observe an F1-score of $0.8515$ under the IID split, however it drops significantly to $0.2423$ on NEAR and to $0.1284$ on FAR splits. This decline highlights the degradation of malware detector over time.

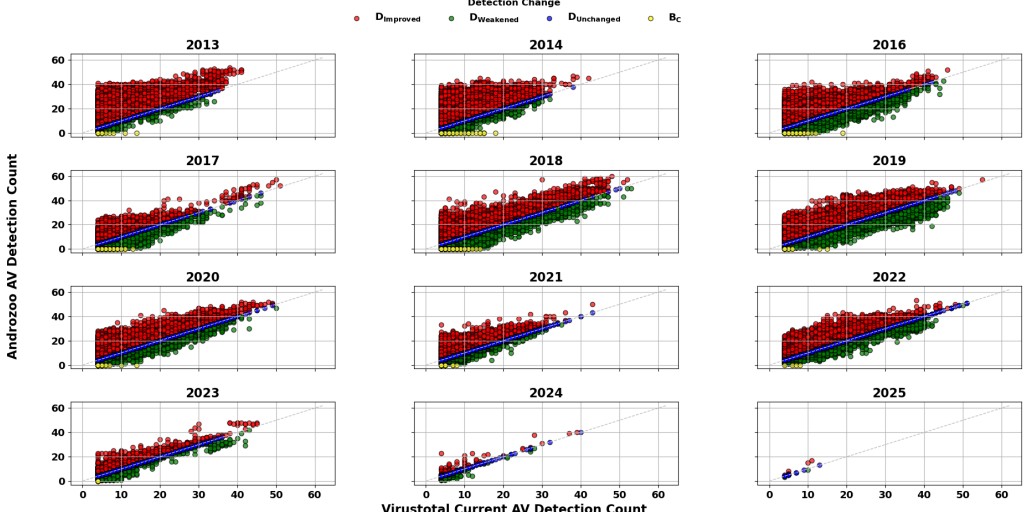

Figure 16: Malware AV detection drift over the years Virustotal vs AndroZoo Metadata. $B_C$: Currently Labeled as Benign, $D_{Improved}$: Improved Detection, $D_{Weakened}$: Weakened Detection, $D_{Unchanged}$: Unchanged Detection.

Figure 15 illustrates the average F1, PR-AUC, and ROC-AUC scores of the baseline malware detectors with varying VT labeling threshold. While Figure 15a is the summarization of Figure 14, subplots (a), (b) and (c) illustrate how performance metrics vary when the VT threshold is set to different values. Across all these three evaluation metrics, we observe only minor differences in baseline malware detectors performance. This suggests that, varying the VT labeling threshold has minimal impact on baseline accuracy. This result suggests that the primary causes of performance degradation in our main experiments may not be the labeling noise from VT, but rather factors such as temporal concept drift and class imbalance. The observed performance degradation is more likely attributed to the distributional shifts over time, reinforcing the relevance of concept drift in real-world malware detection scenarios.

## G   ADDITIONAL VISUALIZATION ON LABEL DRIFT ACROSS YEARS

Figure 16 illustrates the analysis through a visual quantification of label drift across years, using Androozoo metadata and the updated VirusTotal (VT) (VirusTotal, 2025) report.

## H   SCALABILITY OF LAMDA

To support long-term use and extensibility, we have designed LAMDA with scalability in mind. In the context of LAMDA, scalability refers to the extensibility of the dataset—specifically, its ability to be easily expanded with new samples. We have published three variants of LAMDA on the `HuggingFace` repository, each supporting a different `VarianceThreshold` configuration. The dataset creation process begins by splitting the static feature files (i.e., `.data` files extracted from each APK) into stratified train and test splits. From the training split, we collect the global set of all unique tokens (i.e., features), encode both train and test samples into binary vectors in this raw feature space, and apply `VarianceThreshold` to select high-variance features from the training data. The same selected features are then applied to the test data using the saved threshold `object`.

We publish the following artifacts to facilitate scalability: raw feature matrices (before thresholding), reduced feature matrices (after thresholding), and the serialized `VarianceThreshold` object (in `joblib` format). Using these resources and the accompanying codebase, researchers can seamlessly extend LAMDA by collecting newer APKs, extracting static features, encoding them, and applying the same thresholding object to map them into LAMDA's feature space. While it is not feasible to add new samples to the training set—because doing so would alter the global vocabulary

and invalidate the original thresholding, researchers can add test-time samples for evaluation. This supports drift detection on newer and future malware variants without requiring retraining. Thus, LAMDA enables reproducible research and practical testing of detection models against evolving threats.

# I  CONTINUAL LEARNING ON LAMDA

In real-world settings, a large number of new benign and malicious Android applications are introduced each year. As a result, both benign (e.g., due to changes in user demands, Android APIs, security practices) and malicious (e.g., the emergence of novel malware variants) behaviors evolve over time, leading to concept drift. This makes it challenging for static machine learning models to maintain reliable performance over time without retraining regularly. However, complete retraining of past data becomes impractical due to the massive volume of Android applications released daily and the high computational cost associated with retraining. On the other hand, training solely on recent data often leads to catastrophic forgetting (De Lange et al., 2021; Rahman et al., 2022), where previously acquired knowledge is overwritten or lost. In such a situation, continual learning (CL) offers a compelling solution by enabling the detection models to adapt incrementally to new benign and malware applications without the need to retrain with all past data (Park et al., 2025; Rahman et al., 2022; Ghiani et al., 2025). However, some CL techniques may require access to a small subset of past data.

## I.1  LAMDA FOR CONTINUAL LEARNING

LAMDA can be a natural choice for benchmarking CL due to several key properties of its design and structure:

I. **Temporal granularity:** It spans over a decade (2013–2025, excluding 2015) with available both monthly and yearly splits, allowing custom CL as per need.

II. **Concept drift:** As shown in Section 4, LAMDA exhibits significant distributional changes over time, both in feature and label space.

III. **Flexible task construction:**

– **Domain-IL:** Using yearly data splits while maintaining a consistent malware or benign labeling.

– **Class-IL:** Leveraging AVClass2-labeled malware families to incrementally expand the label space.

IV. **Real world relevance:** LAMDA is derived from real-world Android APKs and VirusTotal reports, introducing authentic drift and noise.

We evaluate CL on the LAMDA benchmark using two established baselines, *Naive* (i.e., None) and *Joint*, inspired by the prior work (Rahman et al., 2022; Ghiani et al., 2025). Additionally, we include *Replay* (i.e., Experience Replay) (Rolnick et al., 2019), a state-of-the-art memory replay based CL method, configured with a buffer size of 200 samples per experience. The Naive baseline trains the model sequentially on each experience or task without any access to past data, while the Joint baseline retrains the model from scratch using the cumulative data observed up to the current experience or task.

These baselines are tested under two settings: Domain Incremental Learning (*Domain-IL*), which involves binary malware *vs* benign classification across yearly tasks, and Class Incremental Learning (*Class-IL*), where each task introduces new malware families to classify (van de Ven et al., 2022; Rahman et al., 2022). Due to the lack of available prior work that can assign a single behavioral label, we didn't consider Task Incremental Learning (*Task-IL*) in our experimental setups.

We define each experience or task in the Domain-IL experiment as all samples (both benign and malicious) collected within a specific calendar year (e.g., 2013, 2014, ..., 2025). However, for the Class-IL experiments, each experience or task consists of only the malware samples collected during the corresponding year.

## I.2 Continual Learning Experimental Setup

**Domain-IL.** In this setting, each experience or task corresponds to samples collected during a specific year (i.e., 2013, 2014, ... and so on). The model is designed to continuously learn to distinguish between malware and benign samples as the data distribution evolves over time. We use the `Baseline` variant of our published dataset, treating each year as a separate task in the learning sequence. The objective is for the model to adapt and maintain accurate binary classification performance despite the temporal distribution shifts.

**Class-IL.** In this setting, we utilize a different dataset derived from the `Baseline` variant of our published dataset. We selected only those malware families that contained more than 10 samples in the test set, resulting in a total of 154 families for our experiment. Consequently, we excluded the year 2025 from our experiments, as no family in that split met the minimum sample threshold. Additionally, we omit standard class incremental learning (Park et al., 2025; Rahman et al., 2022), where entirely new classes are introduced in each experience. This approach does not reflect how malware appears in real-world scenarios, malicious samples often come from a mix of previously seen and new families. This claim is supported by the analysis presented in Table 6. The model is expected to learn incrementally to classify samples across all malware families encountered.

**Model Architecture.** We use a shared base architecture, a multi-layer perceptron (MLP) for both Class-IL and Domain-IL settings, consisting of four hidden layers — 512, 384, 256, 128, with ReLU activation. However, Task-specific heads are added to support each learning scenario. For Class-IL, we add a single linear layer outputting logits for all classes and train with categorical cross-entropy loss. For Domain-IL, we use a two-layer MLP head (100 units each, with dropout p=0.2) and a final sigmoid output, trained with binary cross-entropy. All networks are optimized with SGD (learning-rate 0.01, momentum 0.9, weight-decay 0.000001).

**Evaluation Metrics.** We evaluate classification performance using F1 score, which is the harmonic mean of precision and recall. It provides a balanced measure of a model's predictions, particularly important in *imbalanced* datasets. Following the prior work (Ghiani et al., 2025), and we compute the F1 score after training on the $k$-th experience using two complementary evaluation modes:

- **Backward Transfer Performance**: We measures the model's ability to retain knowledge from previous tasks. After training on experience $k$, we compute the F1 scores on all previously seen experiences ($\leq k$). This helps quantify the extent of *catastrophic forgetting* (CF).
- **Forward Transfer Performance**: We measures the model's ability to generalize to future, unseen tasks. After training on experience $n$, we compute the F1 score on all future experiences ($> k$). This indicates how well the model's current knowledge transfers to upcoming distributions.

## I.3 Continual Learning Experimental Results

Figures 17, 18, 19, and 20 demonstrate the effectiveness of the CL methods in evaluating in realistic scenarios using LAMDA benchmark. In the Class-IL setting, we observe strong signs of catastrophic forgetting, especially in the Naive and Replay (*Experience Replay*) strategies. Backward F1 scores drop sharply after certain years, showing that learning new classes without retaining the previous knowledge leads to forgetting. Joint retains high performance as expected due to its exposure to all the previous data. In the Domain-IL setting, we observe that forgetting is relatively limited due to the fixed set of classes (*malware* or *benign*). Although the data distribution evolves over time, which leads to all strategies experiencing a gradual decline in forward F1 scores as they fail to adapt to new distributions. Additionally, we report the average F1 scores of LAMDA across all tasks under the Class-IL and Domain-IL scenarios in Tables 17 and 16. In the Class-IL setting (Table 17), the Joint strategy consistently achieves the highest performance, as expected, due to its access to the full dataset during training. However, this advantage also implies the need for significantly higher computational resources which makes it less practical for real-world settings. The Naive and Replay strategies perform considerably worse, which was also expected as the continual introduction of

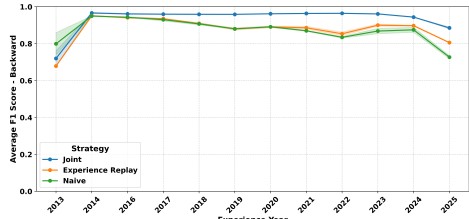

Figure 17: F1 Score in Domain-IL (Forward)

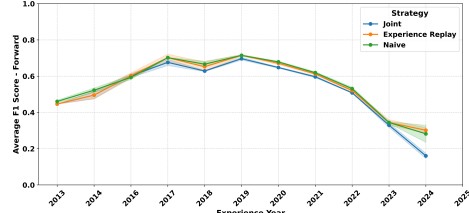

Figure 18: F1 Score in Domain-IL (Backward)

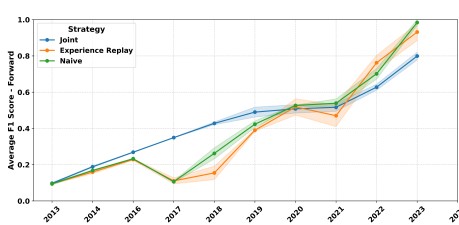

Figure 19: F1 Score in Class-IL (Forward)

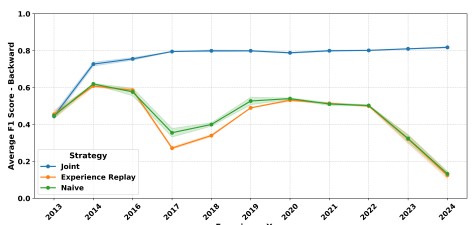

Figure 20: F1 Score in Class-IL (Backward)

new classes. In contrast, the Domain-IL results (Table 16) show generally higher and more stable F1 scores across all strategies. Since the label space remains fixed over time, both Replay and even the Naive strategy perform reasonably well. This observation suggests that the primary challenge in Domain-IL is not always forgetting, but rather adapting to distributional shifts in the data.

These results highlight LAMDA's ability to capture both key challenges in continual learning: class expansion and distributional shift. As such, LAMDA serves as a realistic and challenging benchmark that supports future research in continual learning.

Table 16: Average F1 scores of Domain-IL across all experiences or tasks for LAMDA.

| Year | Strategy | Average F1 Score | Year | Strategy | Average F1 Score |
|------|----------|------------------|------|----------|------------------|
| 2013 | Naive | 48.86 ± 1.15 | 2020 | Naive | 80.26 ± 0.27 |
|      | Joint | 46.89 ± 0.33 |      | Joint | 83.08 ± 0.13 |
|      | Replay | 46.56 ± 0.00 |      | Replay | 79.93 ± 0.27 |
| 2014 | Naive | 59.36 ± 0.99 | 2021 | Naive | 78.61 ± 0.33 |
|      | Joint | 57.38 ± 1.67 |      | Joint | 84.12 ± 0.18 |
|      | Replay | 57.13 ± 1.71 |      | Replay | 79.46 ± 0.55 |
| 2016 | Naive | 68.07 ± 0.82 | 2022 | Naive | 75.86 ± 0.48 |
|      | Joint | 69.21 ± 0.78 |      | Joint | 85.02 ± 0.03 |
|      | Replay | 68.75 ± 1.20 |      | Replay | 77.02 ± 0.52 |
| 2017 | Naive | 77.79 ± 0.31 | 2023 | Naive | 78.10 ± 1.28 |
|      | Joint | 77.05 ± 1.13 |      | Joint | 85.59 ± 0.23 |
|      | Replay | 77.93 ± 1.27 |      | Replay | 80.74 ± 0.73 |
| 2018 | Naive | 76.64 ± 1.06 | 2024 | Naive | 82.54 ± 0.81 |
|      | Joint | 76.61 ± 0.15 |      | Joint | 87.86 ± 0.11 |
|      | Replay | 75.94 ± 1.46 |      | Replay | 84.77 ± 0.11 |
| 2019 | Naive | 79.74 ± 0.16 | 2025 | Naive | 72.71 ± 0.83 |
|      | Joint | 82.76 ± 0.41 |      | Joint | 88.52 ± 0.46 |
|      | Replay | 79.72 ± 0.09 |      | Replay | 80.57 ± 0.15 |

## J COMPUTATIONAL RESOURCES FOR LAMDA GENERATION

All dataset processing and experiments for LAMDA were conducted on a high-performance compute server with the following configuration:

- **CPU**: Dual-socket `Intel Xeon Gold 6430` with a total of 128 logical cores (64 physical cores, 2 threads per core).
- **Memory**: 1 TB RAM, with approximately 810 GB available during runtime.

Table 17: Average F1 scores of Class-IL across all experiences or tasks for LAMDA.

| Year | Strategy | Average F1 Score | Year | Strategy | Average F1 Score |
|------|----------|------------------|------|----------|------------------|
| 2013 | Naive | 12.60 ± 0.65 | 2020 | Naive | 53.50 ± 0.18 |
|      | Joint | 12.89 ± 0.37 |      | Joint | 68.63 ± 0.67 |
|      | Replay | 12.75 ± 0.48 |      | Replay | 52.67 ± 1.34 |
| 2014 | Naive | 25.01 ± 0.28 | 2021 | Naive | 51.83 ± 0.87 |
|      | Joint | 28.64 ± 0.49 |      | Joint | 72.23 ± 0.46 |
|      | Replay | 24.07 ± 0.74 |      | Replay | 50.21 ± 1.51 |
| 2016 | Naive | 32.64 ± 0.72 | 2022 | Naive | 53.91 ± 0.42 |
|      | Joint | 40.14 ± 0.22 |      | Joint | 76.98 ± 0.36 |
|      | Replay | 32.60 ± 0.36 |      | Replay | 54.74 ± 0.43 |
| 2017 | Naive | 19.68 ± 1.12 | 2023 | Naive | 38.39 ± 1.84 |
|      | Joint | 51.14 ± 0.09 |      | Joint | 80.88 ± 0.25 |
|      | Replay | 16.92 ± 1.21 |      | Replay | 37.65 ± 2.46 |
| 2018 | Naive | 32.47 ± 2.06 | 2024 | Naive | 13.27 ± 1.33 |
|      | Joint | 59.66 ± 0.31 |      | Joint | 81.79 ± 0.29 |
|      | Replay | 23.86 ± 2.22 |      | Replay | 12.49 ± 1.76 |
| 2019 | Naive | 47.98 ± 1.71 |      |       |      |
|      | Joint | 65.86 ± 1.20 |      |       |      |
|      | Replay | 44.43 ± 0.26 |      |       |      |

- **GPU**: $4\times$ NVIDIA H100 NVL GPUs with 95.8 GB memory per GPU. Experiments were conducted under `CUDA 12.8` and driver version `570.124.06`.

This infrastructure enabled us to efficiently process over 1 million APKs, large-scale temporal benchmarking over 12 years of Android malware data.

# K    DATASET DOCUMENTATION

## K.1    HOSTED URLs

**DOI.** `https://doi.org/10.57967/hf/5563`

**Hugging Face.** `https://huggingface.co/datasets/IQSeC-Lab/LAMDA`.

**Croissant.** `https://huggingface.co/api/datasets/IQSeC-Lab/LAMDA/croissant`

**GitHub Code Access.** `https://github.com/IQSeC-Lab/LAMDA`

**Project Page.** `https://iqsec-lab.github.io/LAMDA/`

## K.2    DATASET CURATION AND PREPROCESSING METHODOLOGY

- **Dataset Construction:** A corpus of over one million Android Package Kits (APKs), spanning the years 2013 to 2025 with the exclusion of 2015, is compiled from the AndroZoo repository Allix et al. (2016); Alecci et al. (2024). A 20% overhead hases is included in the downloading process to account for download and decompilation failures. The collected APKs are systematically organized into year-specific directories, with subdirectories designated for malware (`[year]/malware/`) and benign applications (`[year]/benign/`).

- **Label Assignment:** Binary classification labels is assigned based on the output of Virus-Total (VT) analysis reported in the AndroZoo repository Allix et al. (2016); Alecci et al. (2024):

    - **Benign:** vt_detection $= 0$
    - **Malware:** vt_detection $\geq 4$
    - **Uncertain:** vt_detection $\in [1, 3]$ (discarded)

- **Malware Family Labeling:** AVClass2 Sebastián et al. (2016) is used to standardize malware family labels using VirusTotal reports. Labels are linked to APKs using `SHA256` hashes to support multi-class and temporal malware analysis.

- **Static Feature Extraction based on Drebin:** Each APK is decompiled using `apktool` Brut (2025) to extract static features:
  - From `AndroidManifest.xml`: permissions, components (activities, services, receivers), hardware features, intent filters
  - From `smali` code: restricted/suspicious API calls, hardcoded URLs/IPs

- **Vectorization & Preprocessing:** Extracted features are vectorized into high-dimensional binary vectors using a bag-of-tokens approach. A global vocabulary ($\sim$9.69M tokens) was constructed. Dimensionality was reduced using `VarianceThreshold` (threshold = 0.001), resulting in 4,561 final features.

- **Data Splitting:** Each year's data is split using stratified sampling:
  - **Training:** 80%
  - **Testing:** 20%

  Class balance is maintained within each split.

- **Storage & Format:** Final dataset is saved in both `.npz` (sparse matrix) and `.parquet` (tabular) formats. Each year's folder includes:
  - `X_train.parquet`
  - `X_test.parquet`

  Metadata columns include: `hash`, `label`, `family`, `vt_count`, `year_month`, followed by binary features.

- **Scalability Support:** We have released global vocabulary, selected features, and preprocessing objects (e.g., `VarianceThreshold`) to enable integration with ML pipelines, including Hugging Face.

### K.3 ACCESSIBILITY AND REPRODUCIBILITY

The dataset has been made publicly available on Hugging Face at `https://huggingface.co/datasets/IQSeC-Lab/LAMDA` and has been assigned a permanent Digital Object Identifier (DOI): `https://doi.org/10.57967/hf/5563`. Furthermore, a dedicated GitHub project page has been created at `https://iqsec-lab.github.io/LAMDA/`, which includes detailed instructions and code to reproduce the reported results.

We are committed to the long-term preservation of our dataset through regular checks aimed at identifying and rectifying any data anomalies. Moreover, we are dedicated to the continuous maintenance of this resource by promptly addressing user inquiries and issues, and by releasing updates and enhancements informed by user feedback.

