# OpenReview forum: "LAMDA: A Longitudinal Android Malware Benchmark for Concept Drift Analysis"
_ICLR.cc/2026/Conference — ICLR 2026 Poster_

### Official Review · Reviewer_dzpd · 2025-10-28

**Soundness:** 3
**Presentation:** 3
**Contribution:** 3
**Rating:** 8
**Confidence:** 5

**Summary:**

This paper introduces LAMDA, a large-scale Android malware dataset spanning 12 years (2013–2025, excluding 2015) with over 1 million APK samples, designed specifically to study concept drift in malware detection. The dataset comprises approximately 37% malware samples across 1,380 families and includes static Drebin features. The authors empirically demonstrate performance degradation of standard ML models over time and analyze feature stability, providing a temporal benchmark substantially larger and more diverse than existing datasets. The paper includes comprehensive drift analysis using multiple techniques including Jeffreys divergence, t-SNE visualization, SHAP-based explanations, and label drift analysis.

**Strengths:**

	Scale and Temporal Scope: Over 1 million samples across 12 years with 1,380 families and 150K singleton samples provide unprecedented temporal coverage and diversity for Android malware research. This addresses a genuine gap in existing datasets.
	Comprehensive Drift Analysis: The multi-faceted approach (supervised learning degradation, feature distribution shifts via Jeffreys divergence, feature stability scores, SHAP-based explanation drift, label drift) provides rich evidence for concept drift. The integration of multiple complementary methods strengthens the analysis.
	Reproducibility and Scalability: Publication of feature matrices, variance threshold objects, and code supports reproducibility. The design enables extensibility to new samples, which is valuable for long-term use.
	Rigorous Experimental Validation: The comparison between LAMDA and APIGraph on identical evaluation protocols effectively demonstrates that LAMDA exhibits stronger, more realistic drift. High standard deviations in LAMDA results versus APIGraph's stability support the claim of pronounced drift.

**Weaknesses:**

Unclear Scan Consistency： The paper does not specify whether VirusTotal labels were obtained from single-pass or repeated scans. Since detection outcomes can vary across rescans, this ambiguity may introduce label inconsistency.

Lack of Intra-Sample Drift Analysis： The study analyzes global and family-level drift but does not consider intra-sample temporal variation—how the same APK’s features might change across time. Such analysis could better capture longitudinal behavior shifts.

Static Feature Limitation： LAMDA focuses exclusively on Drebin-style static features. While this ensures comparability, it may overlook runtime or dynamic behaviors that evolve differently, slightly limiting ecological completeness.

**Questions:**

Collection Procedure and Label Stability: During dataset construction, were APKs scanned once or multiple times on VirusTotal? If repeated scans occurred, how were temporal discrepancies in detection counts handled—by selecting the earliest, latest, or majority label? Clarifying this would help assess label stability across the 12-year span.

Intra-Sample Temporal Drift: Has the team examined how features or VirusTotal labels for the same APK hash change across years? This could quantify intra-sample drift and distinguish it from population-level concept drift.

Dynamic and Hybrid Features: Given the exclusive use of static Drebin features, do the authors plan to include dynamic runtime features (e.g., API invocation traces, network behaviors) or hybrid representations in future LAMDA versions? This would enrich longitudinal analysis and reflect real-world adaptive threats.

Temporal Label Validation: Considering that VirusTotal engines evolve over time, did the authors fix specific engine versions or cross-engine consensus thresholds to mitigate version-induced label drift?

---

> ### Author Response · Authors · 2025-11-23
> **Response for dzpd**
>
> We thank the reviewer for the valuable feedback. We have revised the paper addressing the reviewer's concerns. We will appreciate feedback from the reviewer based on this response.
>
> **1. Collection Procedure and Label Stability:**
>
>  We utilized the most recent scan reports provided by AndroZoo for all of the samples. Additionally, we also retrieved VirusTotal reports in a single pass to analyze potential label drift among these samples, comparing the updated labels with the corresponding AndroZoo's recent report. The results of this analysis are presented in Section 5.4, Table 2.
>
> We plan to perform another scan at the end of December to understand the pattern of label drift and will include the updated analysis, if allowed, in the final version of the paper.
>
>
> **2. Intra-Sample Temporal Drift:**
>
> VirusTotal detection counts can change over time for the same APK, and we analyze this behavior in Section 5.4 by tracking how VirusTotal detection counts evolve for identical APK hashes across different years. This includes cases where a sample first appears as “malicious” and is later flagged as “benign”. This analysis directly measures intra-sample label drift, which is separate from population-level concept drift.
>
> Since LAMDA relies on static features extracted from the Android Manifest and decompiled source code, the feature set for a given APK remains constant even if its VirusTotal detection count changes. Therefore, while label drift occurs at the intra-sample level, we do not observe intra-sample feature drift, because the static Manifest content does not change.
>
>
> **3. Dynamic and Hybrid Features:**
> Absolutely, indeed we are planning and actually working to enhance LAMDA with multi-modal features including but not limited to dynamic analysis-based features and graphical features. Stay tuned!
>
>
> **4. Temporal Label Validation:**
>
> Actually, we used the labels provided by AndroZoo. Their scan dates range from 2009 to 2025, depending on the app. Additionally, To the best of our knowledge, VirusTotal does not allow users to query or freeze engines at a particular point in time. Therefore, we did not fix any engine versions. Instead, we re-scanned all malware samples in our dataset using VirusTotal at the time of data collection. We reported the label drift observed between the original AndroZoo labels and our re-scan labels in Section 5.4. For malware family labels, we used AVClass2 [8] which is a well-accepted process in the prior work [2,3].
>
> Furthermore, as we have mentioned, we plan to perform another scan at the end of December to understand the pattern of label drift and will include the updated analysis, if allowed, in the final version of the paper.
>
>
>
> **References:**
>
>
> [1] Allix, Kevin, et al. "AndroZoo: Collecting millions of android apps for the research community." MSR 2016.
>
> [2] Yang, Limin, et al. “CADE: Detecting and Explaining Concept Drift Samples for Security Applications.” USENIX Security 2021.
>
> [3] Chen, Yizheng, et al. “Continuous Learning for Android Malware Detection.” USENIX Security 2023.
>
> [4] Barbero, Federico, et al. "Transcending transcend: Revisiting malware classification in the presence of concept drift." IEEE S&SP 2022.
>
> [5] Pendlebury, Feargus, et al. "TESSERACT: Eliminating experimental bias in malware classification across space and time." USENIX Security 2019.
>
> [6] Zhang, Xiaohan, et al. "Enhancing state-of-the-art classifiers with api semantics to detect evolved android malware." ACM CCS 2020.
>
> [7] Arp, Daniel, et al. "Drebin: Effective and explainable detection of android malware in your pocket." NDSS 2014.
>
> [8] Sebastián, Silvia, and Juan Caballero. "Avclass2: Massive malware tag extraction from av labels." ACSAC 2020.
>
> [9] He, Yiling, et al. "Combating Concept Drift with Explanatory Detection and Adaptation for Android Malware Classification." arXiv 2024.
>
> [10] Park, Jimin, et al. "MalCL: Leveraging gan-based generative replay to combat catastrophic forgetting in malware classification." AAAI 2025.
>
> [11] Mariconti, Enrico, et al. "MAMADROID: Detecting Android Malware by Building Markov Chains of Behavioral Models."  NDSS 2017.
>
> [12] Y. Wu, et al. "MalScan: Fast Market-Wide Mobile Malware Scanning by Social-Network Centrality Analysis," ASE 2019.

---

> > ### Author Response · Authors · 2025-12-01
> > **Request for the response on our rebuttal**
> >
> > Dear Reviewer dzpd,
> >
> > We have put our best efforts into addressing your concerns through this rebuttal. As we are approaching the end of the reviewer-author discussion period, we kindly request you to share your feedback on our responses.

---

### Official Review · Reviewer_GAMZ · 2025-10-29

**Soundness:** 2
**Presentation:** 2
**Contribution:** 2
**Rating:** 6
**Confidence:** 2

**Summary:**

The authors present LAMDA, a malware dataset that spans 12 years and therefore is aimed at capturing classifier drop in performance due to representational drift.

**Strengths:**

This work is in an area that is now not near my current area of research, thus my lower confidence score.

The dataset is large and to the best of my knowledge the longest longitudinal malware dataset collected to date. The analysis is very thorough.

**Weaknesses:**

See questions

**Questions:**

In Figure (2) for LAMDA, could the authors explain why there is a large performance drop in 2017 and 2018?

Drebin-style features are rather old (from 2014), could the authors support the choice for this feature set?

In Table 4 in the appendix, it seems like there are extremely low malware samples from 2023-2025 compared to the train set. This seems to coincide in Figure (2) LAMDA with a very big performance drop. I wonder if the authors have any comments on this? It seems strange to keep especially the years of 2024 and 2025 given the huge class imbalance.

---

> ### Author Response · Authors · 2025-11-23
> **Response for GAMZ**
>
> We thank the reviewer for the valuable feedback. We have revised the paper addressing the reviewer's concerns. We will appreciate feedback from the reviewer based on this response.
>
> **1. Why large performance drop in 2017 and 2018:**
>
> We observe a clear performance drop in 2017 and 2018, and this aligns with multiple independent indicators of strong drift present in those years.
> - First, Figure 3(a) shows a sharp increase in Jeffrey's divergence between 2016 to 2017 and from 2017 to 2018, indicating substantial shifts in static features such as API usage and permissions.
> - Second, feature stability results in Figure 6(a) highlight that features are most unstable in 2017–2018, with larger fluctuations across malware families, suggesting significant behavioral changes.
> -Finally, SHAP-based explanation drift (Figure 7a) also shows a strong drop in 2017 and 2018, confirming abrupt changes in the model’s decision logic during this period.
>
> These findings collectively show that 2017 and 2018 correspond to periods of unusually large distribution shifts in both the feature and label spaces. The performance drop in Figure 2 is consistent with this pronounced drift. LightGBM, MLP, SVM, and XGBoost all exhibit the same dip, indicating that it can be attributed to data drift rather than model choice.
>
> Furthermore, our analysis on label drift in Table 2 shows higher changes in label drift with 5.19% and 3.16% for 2017 and 2018, respectively, which are among the highest in the entire timeline. As such, we hypothesize whether this label drift can be a catalyst for this significant drop in performance.
>
> To validate our hypothesis, we have performed a set of experiments with the updated label (i.e., correcting labels based on the virus total detection count). Our results yield an F1-score of 33.68% (vs. original 30.41%) for 2017 and 30.11% (vs. original 29.78%) for 2018 using ViT model. These results indicate that label drift has a slight impact on this drop in performance.
>
> **2. Why DREBIN feature:**
>
> DREBIN features are still widely used for studying concept drift in Android malware [2, 4, 9]. DREBIN relies on static features from Android Manifest, which includes sensitive API calls, permissions, intent filters, and other key components. The total number of unique DREBIN features in LAMDA exceeds 9.6 million. This feature set captures a wide range of attributes, including requested permissions (e.g., ACCESS_FINE_LOCATION), declared activities and services, broadcast receivers, required hardware components, and intent filters. We kindly refer to Section 3 for details about raw feature extraction and processing. We also showed the static features and their descriptions in Appendix A (Table 7).
> Furthermore, our analysis shows that DREBIN features are sufficient for studying different types of drift such as feature-space drift (Section 5.1), temporal drift (Section 5.2), and SHAP-based explainability (Section 5.3).
>
> We also would like to note that other available Android static features are either not reproducible [6] or very difficult to reproduce [11, 12] with their available codebase.
>
>
> **3. Why low malware samples from 2023-2025:**
>
> - We collected APK samples from the AndroZoo repository, which provides hashes along with VirusTotal detection counts. Following prior work, we label an APK as malware when its VirusTotal detection count (vt_detected) is ≥ 4 [5].
> - When we began the data collection process early in 2025, the AndroZoo repository contained only a small number of malware samples that met our labeling constraint (vt_detected ≥ 4). While malware samples for 2023–2025 are limited, there are enough benign samples available. In practice, the amount of malware that appears in any month or year is irregular and can be naturally imbalanced. To reflect real-world temporal patterns and to maintain the longitudinal structure needed for concept drift analysis, we decided to keep the years 2024 and 2025 in LAMDA.
>
> - Furthermore, we plan to continue the support of the LAMDA project both in Hugging Face and in GitHub and include features of new APKs as they become available. This will allow us to extend the dataset with more analysis and experiments, including multiple feature representations, multi-modal views of concept drift, and dynamic analysis. This ongoing support will make LAMDA even more useful and help the research community for more impactful related future work.
>
>
> **For references, see rebuttal response of dzpd**

---

> > ### Comment · Reviewer_GAMZ · 2025-11-27
> >
> > Thank you for the response! I have read the other rebuttals and I am happy to keep my score. I will keep my low confidence score as unfortunately this paper is far from my current area of research.

---

### Official Review · Reviewer_x7WH · 2025-10-29

**Soundness:** 3
**Presentation:** 3
**Contribution:** 3
**Rating:** 6
**Confidence:** 4

**Summary:**

The authors introduce LAMDA, a temporally diverse dataset meticulously crafted to tackle the challenges of concept drift in Android malware detection. This benchmark spans 12 years and includes over 1 million samples. Additionally, the authors assess state-of-the-art concept drift adaptation methods, revealing their limitations when applied to LAMDA. This highlights the urgent need for more robust approaches in the field.

**Strengths:**

The paper is well-organized.
The research topic is significant.
The experiments are sufficient.

**Weaknesses:**

It lacks a clear comparison with relevant datasets.
It lacks specific guidance for future work.

**Questions:**

1. It is suggested  to conduct a tabular comparison of the existing data to display information such as the number of malware samples, family types, and distribution over the years. This will clarify the innovative aspects of this study.

2. Has the data been deduplicated? Given that many malware samples exhibit identical features at the characteristic level (even if their hash codes differ), it is crucial to know if any deduplication efforts have been made to ensure diversity in software feature.

3. Please provide feasible research directions for future concept drift adaptation work based on the results of your dataset collection. For instance, does the dataset exhibit characteristics that differ from other datasets, making concept drift adaptation more challenging? While it is still recommended to continuously expand the dataset, simply increasing the amount of data is not sufficient for innovation. It is also necessary to introduce fresh perspectives and methodologies.

4. I am particularly curious about the decision to submit this work to ICLR instead of a conference or journal focused on software engineering or security.

---

> ### Author Response · Authors · 2025-11-23
> **Response for x7WH**
>
> We thank the reviewer for the valuable feedback. We have revised the paper addressing the reviewer's concerns. We will appreciate feedback from the reviewer based on this response.
>
> **1. Tabular comparison with existing dataset:**
> | Year | APIGraph Benign | APIGraph Malware | APIGraph Family | APIGraph Singleton | LAMDA Benign | LAMDA Malware | LAMDA Family | LAMDA Singleton |
> |------|------------------|------------------|------------------|---------------------|--------------|---------------|---------------|------------------|
> | 2012 | 27,613 | 3,066 | 104 | 36 | - | - | - | - |
> | 2013 | 43,873 | 4,871 | 172 | 68 | 42,048 | 44,383 | 213 | 1,550 |
> | 2014 | 52,843 | 5,871 | 175 | 55 | 55,427 | 45,756 | 231 | 2,482 |
> | 2015 | 52,173 | 5,797 | 193 | 53 | - | - | - | - |
> | 2016 | 50,859 | 5,651 | 199 | 68 | 64,059 | 45,134 | 375 | 5,861 |
> | 2017 | 24,930 | 2,620 | 147 | 48 | 77,785 | 21,359 | 207 | 9,063 |
> | 2018 | 38,214 | 4,213 | 128 | 45 | 64,942 | 39,350 | 373 | 20,579 |
> | 2019 | - | - | - | - | 49,465 | 41,585 | 635 | 18,916 |
> | 2020 | - | - | - | - | 55,718 | 46,355 | 588 | 30,644 |
> | 2021 | - | - | - | - | 45,528 | 35,627 | 295 | 30,020 |
> | 2022 | - | - | - | - | 44,768 | 41,648 | 651 | 24,927 |
> | 2023 | - | - | - | - | 46,462 | 7,892 | 224 | 5,922 |
> | 2024 | - | - | - | - | 47,633 | 794 | 64 | 626 |
> | 2025 | - | - | - | - | 44,640 | 23 | 8 | 14 |
> | **Total** | **290,505** | **32,089** |  | **373** | **638,475** | **369,906** |  | **150,604** |
>
>
> **2. Deduplication and feature diversity:**
>
>  LAMDA ensures uniqueness at the APK level by using SHA256 hashes provided by AndroZoo. During dataset construction, we verify that no duplicate hashes are included, and AVClass2 is used to standardize malware family labels to avoid repeated or inconsistent label assignments. Because our goal is to build a benchmark for concept drift analysis, we rely on the natural diversity of samples collected across 12 years rather than manually injecting or removing samples.
>  To address the reviewer’s concern about samples with identical DREBIN features despite having different hashes, we note that LAMDA applies a VarianceThreshold filter to remove static features with extremely low variance across the dataset. This process eliminates features that provide little discrimination and helps reduce redundancy in the feature space. Additionally, we release LAMDA with three variants using different variance thresholds (i.e., 0.01, 0.001, 0.0001), which provide alternative feature sets with varying sparsity and discriminatory power. These variants help researchers study concept drift under different levels of feature diversity.
>
> **3. Characteristics of LAMDA:**
>
> **A.** Unique characteristics of LAMDA compared to other datasets
> - First, APIGraph evolves more gradually and is structurally more homogeneous [6], allowing hierarchical contrastive learning to exploit family-level structure. In contrast, LAMDA spans 12 years and exhibits far stronger concept drift, making it more representative of real-world evolution.
> - Second, LAMDA contains an overwhelming number of singleton samples (105,852) lacking any specific family labels, compared to only 373 in APIGraph [6], which diminishes the effectiveness of family-based contrastive objectives. This property undermines Chen-AL’s [3] technique that depends on hierarchical contrastive loss in which family-level structure guides the representation of learning.
> - Finally, LAMDA contains a comparatively greater number of families than other datasets which exhibit both intra- and inter-family drifts. Our evaluation on prior state-of-the-art such as Chen-AL yields an F1 score of 92% on APIGraph, but its performance drops to 45% on LAMDA.
>
> **B.** Future research direction:
>
> - Firstly, a dynamic learning approach is needed, where the model can recognize these patterns and adjust itself over time. Along with semi-supervised learning, concept adjustment techniques [9] and continual learning [10] can also help a model detect malware more effectively under continuous and evolving drifts.
> - Secondly, multi-modal representation learning may help models handle drifts more effectively over time.
> - Finally, techniques that can handle singleton samples and both intra-class and inter-class distribution shifts would be helpful. In addition, methods that can handle label drifts can further improve robustness under the distribution changes observed in LAMDA.
>
> **4. Why ICLR?**
> Our motivation to submit this work to ICLR is two folds:
> - First, malware concept drift detection and adaptation are mostly studied with respect to a ML based malware detection system. As such, the problem is inherently related to machine learning with an application to cybersecurity.
> - Secondly, this work is primarily positioned as a “dataset and benchmark.” ICLR has a dedicated Datasets and Benchmarks track as a primary subcategory, so we believe ICLR is a suitable venue for our work.
>
> **For references, see rebuttal response of dzpd**

---

> ### Author Response · Authors · 2025-12-01
> **Request for the response on our rebuttal**
>
> Dear Reviewer x7WH,
>
> We have put our best efforts into addressing your concerns through this rebuttal. As we are approaching the end of the reviewer-author discussion period, we kindly request you to share your feedback on our responses.

---

### Meta-Review · Area_Chair_Qv7g · 2025-12-31

**Summary:**

This paper introduces LAMDA, a large-scale longitudinal Android malware benchmark designed for concept drift analysis. The dataset spans 12 years and includes over one million samples with extensive malware family coverage, making it substantially larger and more temporally diverse than existing benchmarks. Reviewers generally agree that the problem addressed is important and that LAMDA fills a clear gap in the literature by enabling systematic study of temporal drift in malware detection. The dataset construction is carefully described, and the accompanying analyses demonstrate meaningful performance degradation, feature instability, and explanation drift over time. Overall, the paper is viewed as a solid dataset and benchmark contribution with clear relevance to the machine learning and security communities.

**Reviewer Concerns:**

The reviewers' main concerns centered on the following points:

(1) The positioning of LAMDA relative to existing Android malware datasets and the need for more explicit comparative summaries.

(2) The reliance on static, Drebin-style features, raising questions about their diversity and representativeness for capturing modern malware behaviors.

(3) Labeling consistency and class imbalance in more recent years, particularly due to the limited number of malware samples available.

(4) The absence of a deeper analysis of intra-sample temporal drift and insufficient guidance on how the dataset could inform future research directions.

**Reviewer Scores:**

The reviewer scores are overall positive and show convergence toward acceptance. Reviewer dzpd provided strong accept recommendations (score 8), emphasizing the dataset’s scale, temporal coverage, and comprehensive drift analysis. Reviewers x7WH and GAMZ gave marginally positive scores (both 6), acknowledging the significance of the contribution while noting limitations in feature choice, dataset comparison, and scope. These reviewers did not indicate a change in score after the rebuttal but confirmed that their main concerns had been addressed satisfactorily. Overall, the reviewer assessments support acceptance as a dataset and benchmark contribution.

---

### Decision · Program_Chairs · 2026-01-26

Accept (Poster)